# Self-Supervised Speech Quality Estimation and Enhancement Using Only Clean Speech

**Szu-Wei Fu [1], Kuo-Hsuan Hung [1][\*], Yu Tsao [2], Yu-Chiang Frank Wang [1]**
[1] NVIDIA, [2] Research Center for Information Technology Innovation, Academia Sinica
`szuweif@nvidia.com,d07528023@ntu.edu.tw,`
`yu.tsao@citi.sinica.edu.tw,frankwang@nvidia.com`

## Abstract

Speech quality estimation has recently undergone a paradigm shift from human-hearing expert designs to machine-learning models. However, current models rely mainly on supervised learning, which is time-consuming and expensive for label collection. To solve this problem, we propose VQScore, a self-supervised metric for evaluating speech based on the quantization error of a vector-quantized-variational autoencoder (VQ-VAE). The training of VQ-VAE relies on clean speech; hence, large quantization errors can be expected when the speech is distorted. To further improve correlation with real quality scores, domain knowledge of speech processing is incorporated into the model design. We found that the vector quantization mechanism could also be used for self-supervised speech enhancement (SE) model training. To improve the robustness of the encoder for SE, a novel self-distillation mechanism combined with adversarial training is introduced. In summary, the proposed speech quality estimation method and enhancement models require only clean speech for training without any label requirements. Experimental results show that the proposed VQScore and enhancement model are competitive with supervised baselines. The code and pre-trained models will be released.

## 1 Introduction

Speech quality estimators are important tools in speech-related applications such as text-to-speech, speech enhancement (SE), and speech codecs, etc. A straightforward approach to measure speech quality is through subjective listening tests. During the test, participants are asked to listen to audio samples and provide their judgment (for example, on a 1 to 5 Likert scale). Hence, the mean opinion score (MOS) of an utterance can be obtained by averaging the scores given by different listeners. Although subjective listening tests are generally treated as the "gold standard," such tests are time-consuming and expensive, which restricts their scalability. Therefore, objective metrics have been proposed and applied as surrogates for subjective listening tests.

Objective metrics can be categorized into handcrafted and machine learning-based methods. The handcrafted metrics are typically designed by speech experts. Examples of this approach include the perceptual evaluation of speech quality (PESQ) (Rix et al., 2001), perceptual objective listening quality analysis (POLQA) (Beerends et al., 2013), virtual speech quality objective listener (ViSQOL) (Chinen et al., 2020), short-time objective intelligibility (STOI) (Taal et al., 2011), hearing-aid speech quality index (HASQI) (Kates & Arehart, 2014a), and hearing-aid speech perception index (HASPI) (Kates & Arehart, 2014b), etc. The computation of these methods is mainly based on comparing degraded speech with its clean reference and hence belongs to intrusive metrics. The requirement for clean speech references significantly hinders their application in real-world conditions.

Machine-learning-based methods have been proposed to eliminate the dependence on clean speech references during inference and can be further divided into two categories. The first attempts to non-intrusively estimate the objective scores mentioned above (Fu et al., 2018; Dong & Williamson, 2020; Zezario et al., 2020; Catellier & Voran, 2020; Yu et al., 2021b; Xu et al., 2022; Kumar et al., 2023). However, during training, **noisy/processed and clean speech pairs** are still required to obtain

---

[\*]Internship at NVIDIA

the objective scores as model targets. For example, Quality-Net (Fu et al., 2018) trains a bidirectional long short-term memory (BLSTM) with frame-wise auxiliary loss to predict the PESQ score. Instead of treating it as a regression task, MetricNet (Yu et al., 2021b) estimates the PESQ score using a multi-class classifier trained by the earth mover's distance (Rubner et al., 2000). MOSA-Net (Zezario et al., 2022) is an unified model that simultaneously predict multiple objective scores such as PESQ, STOI, HASQI, and source-to-distortion ratio (SDR). NORESQA (Manocha et al., 2021) utilizes non-matching references to predict relative speech assessment scores (i.e., the signal-to-noise ratio (SNR) and scale-invariant SDR (SI-SDR) (Le Roux et al., 2019)). Although these models release the requirement of corresponding clean reference during inference, their training targets (objective metrics) are generally not perfectly correlated with human judgments (Reddy et al., 2021b).

The second category of machine-learning-based methods (Lo et al., 2019; Leng et al., 2021; Mittag et al., 2021; Tseng et al., 2021; Reddy et al., 2021b; 2022; Tseng et al., 2022; Manocha & Kumar, 2022) has been proposed to solve this problem by using **speech and its subjective scores (e.g., MOS)** for model training. VoiceMOS challenge (Huang et al., 2022) is targeted for automatic prediction of MOS for synthesized speech. DNSMOS (Reddy et al., 2021b; 2022) is trained on large-scale crowdsourced MOS rating data using a multi-stage self-teaching approach. MBNet (Leng et al., 2021) consists of MeanNet, which predicts the mean score of an utterance, and BiasNet, which considers the bias caused by listeners. NORESQA-MOS (Manocha & Kumar, 2022) leverages pre-trained self-supervised models with non-matching references to estimate the MOS and was recently released as a package in TorchAudio (Kumar et al., 2023).

However, to train a robust quality estimator, large-scale listening tests are required to collect paired speech and MOS data for model supervision. For example, the training data for DNSMOS P.835 (Reddy et al., 2022) was 75 h. NORESQA-MOS (Manocha & Kumar, 2022) was trained on 7,000 audio recordings and their corresponding MOS ratings. In addition to the high cost of collecting training data, these supervised models may exhibit poor generalizability to new domains (Maiti et al., 2022). To address this issue, the SpeechLMScore (Maiti et al., 2022), an unsupervised metric for evaluating speech quality using a speech-based language model, was proposed. This metric first maps the input speech to discrete tokens, and then applies a language model to compute its average log-likelihood. Because the language model is trained only on clean speech, a higher likelihood implies better speech quality.

Inspired by SpeechLMScore, we investigated unsupervised speech quality estimation in this study, but used a different approach. Our method was motivated by autoencoder-based anomaly detection (An & Cho, 2015). Because the autoencoder is trained only on normal data, during inference, we expect to obtain a low reconstruction error for normal data and a large error for abnormal data. Kim (2017) applied a speech autoencoder, whose input and output were trained to be as similar as possible if inputs clean speech, to select the most suitable speech enhancement model from a set of candidates. Although (Soni & Patil, 2016; Wang et al., 2019; Martinez et al., 2019) also used autoencoders for speech-quality estimation, the autoencoders in their works were only used for feature extraction. Therefore, a supervised model and quality labels are still **required**. Other works, such as (Giri et al., 2020; Pereira et al., 2021; Ribeiro et al., 2020; Hayashi et al., 2020; Abbasi et al., 2021) mainly applied autoencoders for audio anomaly detection.

Our proposed quality estimator is based on the quantization error of VQ-VAE (Van Den Oord et al., 2017), and we found that VQ-VAE can also be used for self-supervised SE. To align the embedding space, Wang et al. (2020) applied cycle loss to share the latent representation between clean autoencoder and mixture autoencoder. Although paired training data is not required for their model training, noisy speech and noise samples are still **needed**. On the other hand, we achieve representation sharing through the codebook of VQ-VAE. In addition, by the proposed **self-distillation** and **adversarial training**, the enhancement performance can be further improved.

## 2 METHOD

### 2.1 MOTIVATION

As mentioned in the previous section, the proposed speech quality estimator was motivated by autoencoder-based anomaly detection. By measuring the reconstruction error with a suitable threshold, anomalies can be detected even though only normal data are used for model training. In this study,

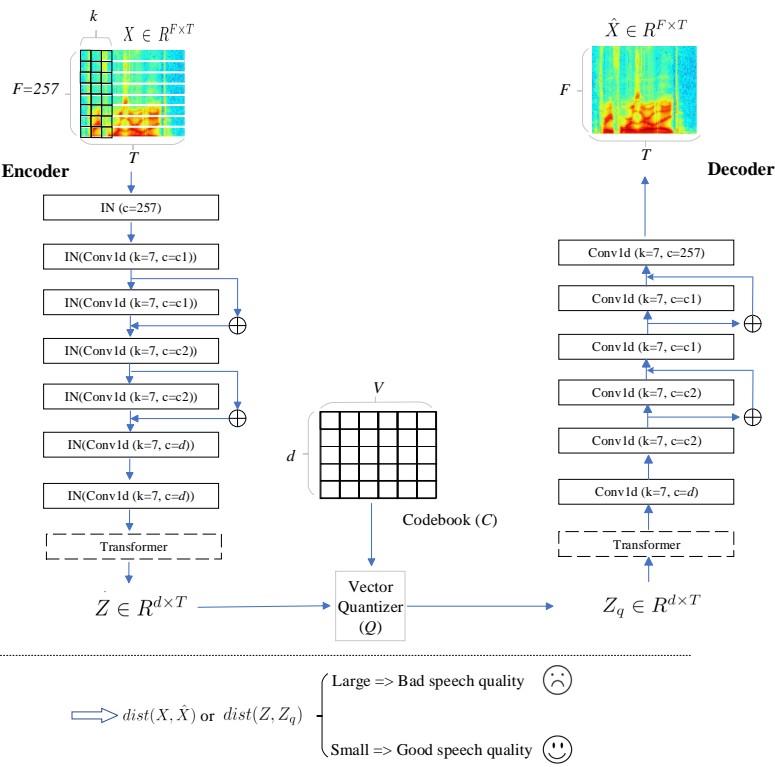

Figure 1: Proposed VQ-VAE for self-supervised speech quality estimation and enhancement. The Transformer blocks are only used for speech enhancement.

we go one step further based on the assumption that the reconstruction error and speech quality may appear in an *inverse proportion* relationship (i.e., a larger reconstruction error may imply lower speech quality). People usually rate the speech quality based on an implicit comparison to what clean speech should sound. **The purpose of training the VQ-VAE with a large amount of clean speech is to guide the model in building its own image of clean speech (stored in the codebook).** In this study, we conducted a comprehensive investigation to address the following questions: 1) Which metric should be used to estimate the reconstruction error? 2) Where should reconstruction error be measured?

While developing a self-supervised speech quality estimator, we also found a potential method for training a speech enhancement model using only clean speech.

## 2.2 PROPOSED MODEL FRAMEWORK

The proposed model comprises three building blocks, as shown in Figure 1.

1) **Encoder** ($E$) maps the input spectrogram $X \in R^{F \times T}$ onto a sequence of embeddings $Z \in R^{d \times T}$, where $F$ and $T$ are the frequency and time dimensions of the spectrogram, respectively, and $d$ is the embedding feature dimension.

We first treat the input spectrogram $X$ as a $T$-length 1-D signal with $F$ channels, and then build the encoder using a series of 1-D convolution layers, as shown in Figure 1. In this figure, $k$ and $c$ represent the kernel size and number of output channels (number of filters), respectively. Note that we apply instance normalization (IN) (Ulyanov et al., 2016) to input $X$ and after every convolutional layer. We found that normalization is a **critical** step for boosting the quality estimation performance. Between the IN and convolution layers, LeakyReLU (Xu et al., 2015) was applied as an activation function.

To increase the model's capacity for speech enhancement, two Transformer encoder layers were inserted before and after the vector quantization module (as indicated by the dashed rectangles in Figure 1), respectively. The standard deviation normalization used in IN is inappropriate for SE because the volume information is also important for signal reconstruction. Therefore, we only maintain the **mean removal** operation in IN for SE.

2) **Vector quantizer** $(Q)$ replaces each embedding $Z_t \in R^{d \times 1}$ with its nearest neighbor in the codebook $C \in R^{d \times V}$, where $t$ is the index along the time dimension and $V$ is the size of the codebook. The codebook is initialized using the k-means algorithm on the first training batch as SoundStream (Zeghidour et al., 2021) and is updated using the exponential moving average (EMA) (Van Den Oord et al., 2017). During inference, the quantized embedding $Z_{q_t} \in R^{d \times 1}$ is chosen from $V$ candidates of the codebook, such that it has the smallest $L_2$ distance:

$$Z_{q_t} = \arg\min_{C_v \in C} ||Z_t - C_v||_2 \tag{1}$$

We can also normalize the embedding and codebook to have unit $L_2$ norm before calculating the $L_2$ distance:

$$Z_{q_t} = \arg\min_{C_v \in C} ||norm_{L_2}(Z_t) - norm_{L_2}(C_v)||_2 \tag{2}$$

This is equivalent to choosing the quantized embedding based on cosine similarity (Yu et al., 2021a; Chiu et al., 2022). These two criteria have their own applications. For example, Eq. (1) is suitable for speech enhancement while Eq. (2) is good at modeling speech quality. We will discuss details in the following sections.

3) **Decoder** $(D)$, which generates a reconstruction of input $\hat{X} \in R^{F \times T}$ from quantized embeddings $Z_q \in R^{d \times T}$. The decoder architecture is similar to the encoder as shown in Figure 1.

## 2.3 TRAINING OBJECTIVE

The training loss of the VQ-VAE includes three loss terms (Van Den Oord et al., 2017):

$$L = dist(X, \hat{X}) + ||sg(Z_t) - Z_{q_t}||_2 + \beta||Z_t - sg(Z_{q_t})||_2 \tag{3}$$

where $sg(.)$ represents the stop-gradient operator. The second term is used to update the codebook, where, in practice, the EMA is applied. The third term is the commitment loss, which causes the encoder to commit to the codebook. In this study, the commitment weight $\beta$ was set to 1.0 and 3.0 for quality estimation and SE, respectively, based on the performance on validation set. The first term is the reconstruction loss of the input $X$ and output $\hat{X}$. Conventionally, this is simply an $L_1$ or $L_2$ loss. However, for speech quality estimation, we applied negative cosine similarity as the distance metric.

The reason for using cosine similarity in Eqs. (2) and (3) for quality estimation is that we want similar phonemes can be grouped in the same token of the codebook. **Using cosine similarity can ignore the volume difference and focus more on the content.** For example, if we apply $L_2$ loss to minimize the reconstruction loss, louder and quieter 'a' sounds may not be grouped into the same code, which hinders the evaluation of speech quality.

## 2.4 VQSCORE FOR SPEECH QUALITY ESTIMATION

In conventional autoencoder-based anomaly detection, the criterion for determining an anomaly is based on the reconstruction errors of the model input and output. In this study, we found that the quantization error between $Z$ and $Z_q$ can provide a much higher correlation with **human hearing perception**. Note that being able to calculate the quantization error is a unique property of the VQ-VAE that other autoencoders do not have. Because the VQ-VAE was trained only with clean speech, its codebook can be treated as a high-level representation (e.g., phonemes) for speech signals. Therefore, the similarity calculated in this space aligns better with subjective quality scores. The proposed VQScore is hence defined as:

$$VQScore_{(cos,z)}(X) = \frac{1}{T} \sum_{t=1}^{T} cos(Z_t, Z_{q_t}) \tag{4}$$

where $cos(.)$ is cosine similarity. $(cos, z)$ in $VQScore_{(cos,z)}$ represents using the cosine similarity as the distance metric and it is calculated in code space ($z$). We compare the performances of different combinations (e.g., $VQScore_{(cos,x)}$ and $VQScore_{(L_2,x)}$, etc.) in the Section B of Appendix.

## 2.5 Self-Distillation with Adversarial Training to Improve Model Robustness for Speech Enhancement

The robustness of the encoder to out-of-domain data (i.e., noisy speech) is the key to self-supervised SE. **Once the encoder can map the noisy speech to the corresponding tokens of clean speech, or the decoder has the error correction ability, speech enhancement can be achieved.** Based on this observation, our proposed self-supervised SE model training contains 2 steps.

**Step 1 (VQ-VAE training)**: Train a normal VQ-VAE using Eq. (3) with clean speech. After the VQ-VAE training converges, it will be served as a teacher model $T$. In addition, initialize a student model $S$ from the weights of the teacher model. The self-distillation (Zhang et al., 2019) mechanism will be used for the next training step.

**Step 2-1 (Adversarial attack)**: To further improve the robustness of the student model, its encoder and decoder are fine-tuned by adversarial training (AT) (Goodfellow et al., 2014; Bai et al., 2021) with its codebook being fixed.

Instead of adding some predefined noise that follows a certain probability distribution (e.g., Gaussian noise) to the input clean speech, the adversarial noise is applied, which is the most confusing noise to the model for making incorrect token predictions. Given a clean speech $X$ and the encoder of the teacher model $T_{enc}$, its quantized token $Z_{T_q}$ can be obtained using Eq. (1). The adversarial noise $\delta$ of the encoder of the student model $S_{enc}$ can be found by solving the following optimization problem:

$$\max_{\delta} L_{ce}(S_{enc}(X + \delta), Z_{T_q}|C) \tag{5}$$

Because the token selection is based on the distance between the encoder output and the candidates in the dictionary $C$, (i.e., Eq. (1)), we can formulate this process as a probability distribution based on the distance and softmax operation (e.g., if the distance is smaller, it is more likely to be chosen). Therefore, the cross-entropy loss $L_{ce}$ in Eq. (5) can be calculated as :

$$L_{ce} = -\frac{1}{T} \sum_{t=1}^{T} log(\frac{exp(-||(S_{enc}(X + \delta)_t - Z_{T_{qt}}||_2)}{\sum_{v=1}^{V} exp(-||(S_{enc}(X + \delta)_t - C_v||_2)}) \tag{6}$$

The obtained noise $\delta$ when adding to the clean speech $X$, will **maximize** the cross-entropy loss between tokens from the student and teacher model.

**Step 2-2 (Adversarial training)**: To improve the robustness of the encoder part of the student model, the adversarial attacked input $X + \delta$ will be fed into the student model and the weights are updated to **minimize** the cross-entropy loss between its token predictions and the ground truth tokens provided by the teacher model (with clean speech as input) using the following loss function:

$$\min_{S_{enc}} L_{ce}(S_{enc}(X + \delta), Z_{T_q}|C) \tag{7}$$

In addition, to obtain a robust decoder of the student model, an $L_1$ loss between clean speech and the decoder output (with adversarial attacked tokens as inputs) is also applied. Experimental results show that this will slightly improve the overall performance.

Steps 2-1 and 2-2 will be alternatively applied, and the student model serves as the final SE model.

## 3 Experiments

### 3.1 Test Sets and Baselines for Speech Quality Estimation

The test set used for the speech quality estimation was obtained from the Conferencing Speech 2022 Challenge (Yi et al., 2022). First, we randomly sampled 200 clips from IU Bloomington COSINE

(IUB_cosine) (Stupakov et al., 2009) and VOiCES (IUB_voices) (Richey et al., 2018), individually. For the VOiCES dataset, acoustic conditions such as foreground speech (reference), low-pass filtered reference (anchor), and reverberants were included. For the COSINE dataset, close-talking mic (reference) and chest or shoulder mic (noisy) data were provided. The second source of the test set was the Tencent corpus, which included Chinese speech with (Tencent_wR) and without (Tencent_woR) reverberation. In the without-reverberation condition, speech clips were artificially added with some damage to simulate a scenario that may be encountered in an online meeting (e.g., background noise, high-pass/low-pass filtering, amplitude clipping, codec processing, noise suppression, and packet loss concealment). In the reverberation condition, simulated reverberation and speech recorded in a realistic reverberant room were provided. We randomly sampled 250 clips from each subset. A list of sampled clips will be released to facilitate model comparison. The VoiceBank-DEMAND noisy test set (Valentini-Botinhao et al., 2016) was selected as the validation set. Because it does not come with quality labels, we set the training stop point when the VQScore reached the highest correlation with its DNSMOS P.835 (OVRL) (Reddy et al., 2022).

Two supervised quality estimators, DNSMOS P.835 and TorchaudioSquim (MOS) (Kumar et al., 2023), were used for baseline comparison. DNSMOS P.835 provided three audio scores: speech quality (SIG), background noise quality (BAK), and overall quality (OVRL). OVRL was selected as the baseline because it had a higher correlation with the real quality scores in our preliminary experiments. The SpeechLMScore was selected as the baseline for the self-supervised method.

## 3.2 Experimental Results of Speech Quality Estimation

The training data used to train our VQ-VAE for quality estimation was the LibriSpeech clean 460 hours (Panayotov et al., 2015). The model structure is shown in Figure 1, where the codebook size $V$ and code dimension $d$ are set to (2048, 32) and $(c_1, c_2)$=(128, 64). We first calculated the conventional objective quality metrics (i.e., SNR, PESQ, and STOI) and DNSMOS P.835 on the validation set (VoiceBank-DEMAND noisy test set). We then calculated the linear correlation coefficient (Pearson, 1920) between those scores with SpeechLMScore and the proposed VQScore. The experimental results are presented in Table 1. From this table, we can observe that, for metrics related to human perception, the VQScore calculated in the code space ($z$) generally performs much better than that calculated in the signal space ($x$). Our VQScore$_{(cos,z)}$ had a very high correlation with DNSMOS P.835 (BAK), implying that it has a superior ability to detect noise. It also outperformed SpeechLMScore across all metrics.

Next, as shown in Table 2, we compare the correlation between different quality estimators and real quality scores on the test set (the scatter plots are shown in Section C of Appendix). TorchaudioSquim (MOS) did not perform as well as DNSMOS, possibly due to limited training data and domain mismatch (its training data, BVCC (Cooper & Yamagishi, 2021) came from text-to-speech and the voice conversion challenge). In contrast, the proposed VQScore was competitive with DNSMOS, although **NO** quality labels were required during training. The VQScore also outperformed SpeechLMScore, possibly because the SpeechLMScore is based on the perplexity of the language model, so minor degradation or noise may not change the output of the tokenizer and the following language model. Note that the training data of our VQScore is only based on LibriSpeech clean 460 hours which is a **subset** (roughly 460/(960+5,600) ≈ 7%) used to train SpeechLMScore. The proposed framework can also be used for frame-level SNR estimation as discussed in the section D of Appendix.

## 3.3 Test Sets and Baselines for Speech Enhancement

The test sets used for evaluating different speech enhancement models came from three sets: the VoiceBank-DEMAND noisy test set, DNS1 test set (Reddy et al., 2020), and DNS3 test set (Reddy et al., 2021a).

The VoiceBank-DEMAND noisy test set is a relatively simple dataset for SE because only two speakers and additive noise are included. In contrast, the blind test set in DNS1 covers different acoustic conditions, such as noisy speech without reverberation (noreverb), noisy reverberant speech (reverb), and noisy real recordings (real). The DNS3 test set can be divided into subcategories based on realness (real or synthetic) and language (English or non-English).

Table 1: Linear correlation coefficient between different objective metrics and the proposed VQScore on the VoiceBank-DEMAND noisy test set (Valentini-Botinhao et al., 2016). For metrics related to human perception, $\text{VQScore}_{(cos,z)}$ performs much better than $\text{VQScore}_{(cos,x)}$.

|  | SpeechLMScore (Maiti et al., 2022) | $\text{VQScore}_{(cos,x)}$ | $\text{VQScore}_{(cos,z)}$ |
|---|---|---|---|
| SNR | 0.4806 | 0.5177 | **0.5327** |
| PESQ | 0.5940 | 0.6514 | **0.7941** |
| STOI | 0.6023 | 0.5451 | **0.7490** |
| DNSMOS P.835 (SIG) | 0.5310 | 0.4051 | **0.5620** |
| DNSMOS P.835 (BAK) | 0.7106 | 0.6836 | **0.8773** |
| DNSMOS P.835 (OVR) | 0.7045 | 0.6370 | **0.8386** |

Table 2: Linear correlation coefficient between **real** quality scores and different quality estimators on different test sets.

|  | Supervised training | | Self-Supervised training | |
|---|---|---|---|---|
|  | DNSMOS P.835 (Reddy et al., 2022) | TorchaudioSquim (Kumar et al., 2023) | SpeechLMScore (Maiti et al., 2022) | $\text{VQScore}_{(cos,z)}$ |
| Tencent_wR | **0.6566** | 0.4040 | 0.5910 | 0.5865 |
| Tencent_woR | **0.7769** | 0.5025 | 0.7079 | 0.7159 |
| IUB_cosine | 0.3938 | 0.3991 | 0.3913 | **0.4880** |
| IUB_voices | 0.8181 | 0.6984 | 0.6891 | **0.8604** |

For comparison with our self-supervised SE model, two signal-processing-based methods, MMSE (Ephraim & Malah, 1984) and Wiener filter (Loizou, 2013) were included as baselines. Noisy-target training (NyTT) (Fujimura et al., 2021) and MetricGAN-U (Fu et al., 2022) are two approaches that are different from conventional supervised SE model training. NyTT does not need noisy and clean training pairs, it creates training pairs by adding noise to noisy speech. The noise-added signal and original noisy speech are used as the model input and target, respectively. MetricGAN-U is trained on noisy speech with the loss from the DNSMOS model (which actually requires extra (speech, MOS) pairs data for training). For the supervised baselines, the first one is CNN-Transformer, which has the same model structure as our self-supervised-based model except for the removal of the VQ module. Another model that achieved good results on the VoiceBank-DEMAND noisy test set was also selected: Demucs (Defossez et al., 2020) is an SE model that operates in the waveform domain. Our self-supervised-based SE model is VQ-VAE trained only on clean speech with $(V, d, c_1, c_2)$=(4096, 128, 200, 150).

## 3.4 EXPERIMENTAL RESULTS OF SPEECH ENHANCEMENT

### 3.4.1 SPEECH ENHANCEMENT RESULTS OF MATCHED AND MISMATCHED CONDITIONS

To make a fair comparison with the supervised baselines, we provide the results of our self-supervised SE model trained only on the clean speech of the VoiceBank-DEMAND training set (i.e., the corresponding noisy speech is **NOT** used during our model training). In Table 3, we present the results for the matched condition, in which the training and evaluation sets were from the same source. Compared with noisy speech and traditional signal-processing-based methods, Proposed + AT showed a significant improvement in SIG, BAK, and OVRL. As expected, the effect of AT is

---

[1]Note that several recent papers have shown that PESQ can not reflect the true speech quality, especially for speech generated by a generative model (such as GAN Kumar et al. (2020); Liu et al. (2022), vocoder Maiti & Mandel (2020); Li & Yamagishi (2020); Du et al. (2020), diffusion model Serrà et al. (2022), etc.) We also observe that the PESQ scores of enhanced speech generated by models with VQ usually **CANNOT** reflect the true speech quality. This is mainly because the discrete tokens in VQ are shared by similar sounds, which makes the generated speech have less fidelity. However, as pointed out by DNSMOS and the following subjective listening test, this does not imply its generated speech has lower quality.

[2]Comes from several different data sources.

Table 3: Comparison of different SE models on the VoiceBank-DEMAND noisy test set. Training data comes from the training set of VoiceBank-DEMAND. The underlined numbers represent the best results for the supervised models. The bold numbers represent the best results for the models that do not need (noisy, clean) training data pairs.

| Model | Training data | PESQ[1] | SIG | BAK | OVRL |
|---|---|---|---|---|---|
| Clean | - | 4.64 | 3.463 | 3.961 | 3.152 |
| Noisy | - | 1.97 | 3.273 | 2.862 | 2.524 |
| CNN-Transformer | (noisy, clean) pairs | 2.79 | 3.389 | 3.927 | 3.070 |
| Demucs (Defossez et al., 2020) | (noisy, clean) pairs | 2.95 | 3.436 | 3.951 | 3.123 |
| NyTT (Fujimura et al., 2021) | (noisy speech, noise) [2] | 2.30 | **3.444** | 3.106 | 2.736 |
| MetricGAN-U (Fu et al., 2022) | noisy speech + DNSMOS model | 2.13 | 3.200 | 3.400 | 2.660 |
| MMSE (Ephraim & Malah, 1984) | - | 2.19 | 3.215 | 3.089 | 2.566 |
| Wiener (Loizou, 2013) | - | 2.23 | 3.208 | 2.983 | 2.501 |
| Proposed | clean speech | 2.20 | 3.329 | 3.646 | 2.876 |
| Proposed + AT | clean speech | **2.38** | 3.300 | **3.838** | **2.941** |

mainly to make the encoder more robust to noise, and hence boost the model's denoise ability (PESQ and BAK improve by 0.18 and 0.192, respectively). Compared to NyTT and MetricGAN-U, our Proposed + AT has significant improvement in PESQ, BAK, and OVRL. Both CNN-Transformer and Demucs can generate speech with good quality (in terms of the DNSMOS scores) under this matched condition.

To evaluate the model's generalization ability, we compared their performance under mismatched conditions, where the training and testing sets originated from different sources. Table 4 lists the results for the DNS1 test set. Although the performance of CNN-Transformer is worse than Demucs in the matched condition (Table 3), it generally performs better in this mismatched condition. In addition, it can be observed that even though adding adversarial noise or Gaussian noise on the clean input for self-supervised model training can further improve the scores of all the evaluation metrics, the improvement from adversarial noise was more prominent. For the OVRL scores, Proposed + AT was competitive with the supervised CNN-Transformer, and outperformed Demucs, especially in the more mismatched cases (Real and Reverb cases). The experimental results for DNS3 are presented in Section E of Appendix. The same trend appeared as in the case of DNS1: the proposed model with AT can significantly outperform Demucs in the more mismatched cases (i.e., Real and non-English).

### 3.4.2 RESULTS OF LISTENING TEST

Since objective evaluation metrics may not consistently capture the genuine perceptual experience, we conducted a subjective listening test. In order to assess the subjective perception, we compare our Proposed + AT with noisy, Wiener, CNN-Transformer, and Demucs. For each acoustic condition (**real**, **noreverb**, and **reverb**), 8 samples were randomly selected from the test set, amounting to a total of $8 \times 5$ (different enhancement methods and noisy) $\times 3$ (acoustic conditions) = 120 utterances that each listener was required to evaluate. For each signal, the listener rated the speech quality ($SIG_{sub}$), background noise removal ($BAK_{sub}$), and the overall quality ($OVRL_{sub}$) follows ITU-T P.835. 17 listeners participated in the study. Table 5 shows the results of the listening test. It can be observed that in every scenario, our Proposed + AT exhibits the best noise reduction capability (highest $BAK_{sub}$ score). On the other hand, our method has larger speech distortion (lower $SIG_{sub}$ score) compared to the CNN-Transformer, which has the same model structure but is trained in a supervised way. In terms of $OVRL_{sub}$, our Proposed + AT is competitive with the CNN-Transformer and outperforms other baselines. These results verify that our self-supervised model has better generalization capability than Demucs and is comparable to CNN-Transformer.

## 4 CONCLUSION

In this study, we propose a novel self-supervised speech quality estimator trained only on clean speech. Motivated by anomaly detection, if the input speech has a different pattern from that of the

Table 4: Comparison of different SE models on the DNS1 test set. Training data comes from the training set of VoiceBank-DEMAND.

| Subset | Model | Training data | SIG | BAK | OVRL |
|---|---|---|---|---|---|
| Real | Noisy | - | 3.173 | 2.367 | 2.238 |
| | CNN-Transformer | (noisy, clean) pairs | 3.074 | 3.339 | 2.620 |
| | Demucs (Defossez et al., 2020) | (noisy, clean) pairs | 3.073 | 3.335 | 2.570 |
| | Wiener (Loizou, 2013) | - | **3.207** | 2.579 | 2.313 |
| | Proposed | clean speech | 3.095 | 3.365 | 2.589 |
| | Proposed + Gaussian | clean speech | 3.152 | 3.458 | 2.673 |
| | Proposed + AT | clean speech | 3.156 | **3.640** | **2.750** |
| Noreverb | Noisy | - | 3.492 | 2.577 | 2.513 |
| | CNN-Transformer | (noisy, clean) pairs | 3.515 | 3.786 | 3.124 |
| | Demucs (Defossez et al., 2020) | (noisy, clean) pairs | **3.535** | 3.651 | 3.073 |
| | Wiener (Loizou, 2013) | - | 3.311 | 2.747 | 2.447 |
| | Proposed | clean speech | 3.463 | 3.764 | 3.066 |
| | Proposed + Gaussian | clean speech | 3.484 | 3.830 | 3.115 |
| | Proposed + AT | clean speech | 3.481 | **3.960** | **3.162** |
| Reverb | Noisy | - | 2.057 | 1.576 | 1.504 |
| | CNN-Transformer | (noisy, clean) pairs | 2.849 | 3.352 | 2.409 |
| | Demucs (Defossez et al., 2020) | (noisy, clean) pairs | 2.586 | 3.260 | 2.175 |
| | Wiener (Loizou, 2013) | - | 2.649 | 2.251 | 1.838 |
| | Proposed | clean speech | 2.911 | 3.097 | 2.325 |
| | Proposed + Gaussian | clean speech | 2.930 | 3.222 | 2.394 |
| | Proposed + AT | clean speech | **2.949** | **3.361** | **2.456** |

Table 5: Listening test results of different SE models on the DNS1 test set.

| Subset | Model | $SIG_{sub}$ | $BAK_{sub}$ | $OVRL_{sub}$ |
|---|---|---|---|---|
| Real | Noisy | **3.890** | 2.294 | 2.809 |
| | CNN-Transformer | 3.537 | 2.801 | **3.044** |
| | Demucs (Defossez et al., 2020) | 2.890 | 2.515 | 2.515 |
| | Wiener (Loizou, 2013) | 3.787 | 2.250 | 2.868 |
| | Proposed + AT | 3.272 | **2.978** | 3.000 |
| Noreverb | Noisy | **3.765** | 2.059 | 2.647 |
| | CNN-Transformer | 3.706 | 2.809 | 3.088 |
| | Demucs (Defossez et al., 2020) | 3.676 | 2.779 | 3.051 |
| | Wiener (Loizou, 2013) | 3.404 | 2.147 | 2.654 |
| | Proposed + AT | 3.404 | **3.162** | **3.132** |
| Reverb | Noisy | **3.169** | 1.691 | 2.176 |
| | CNN-Transformer | 2.610 | 2.632 | **2.382** |
| | Demucs (Defossez et al., 2020) | 1.588 | 1.934 | 1.515 |
| | Wiener (Loizou, 2013) | 2.963 | 2.015 | 2.250 |
| | Proposed + AT | 2.522 | **2.721** | **2.382** |

clean speech, the reconstruction error may be larger. Instead of directly computing the error in the signal domain, we find that it can provide a higher correlation with other objective and subjective scores when the distance is calculated in the code space (i.e., the quantization error) of the VQ-VAE. Although no quality labels are required during model training, the correlation coefficient between the real quality scores and the proposed VQScore is competitive with that of the supervised estimators. Next, under the VQ-VAE framework, the key to self-supervised speech enhancement is the robustness of the encoder and decoder. Therefore, a novel self-distillation mechanism combined with adversarial training is proposed which can achieve good SE results without the need for any (noisy, clean) speech training pairs. Both the objective and subjective experimental results show that the proposed self-supervised framework is competitive with that of supervised SE models under mismatch conditions.

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

# Appendix

## A    LEARNING CURVES ON VOICEBANK-DEMAND NOISY TEST SET FOR SPEECH QUALITY ESTIMATION

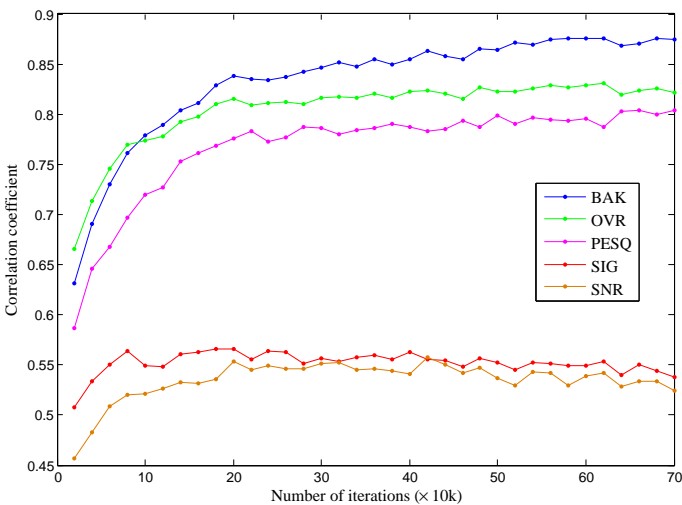

Figure 2: Learning curves of the correlation coefficient between various objective metrics and the proposed $\text{VQScore}_{(cos,z)}$ on the VoiceBank-DEMAND noisy test set (Valentini-Botinhao et al., 2016).

Figure 2 presents the learning curves of the correlation coefficient between various objective metrics and the proposed $\text{VQScore}_{(cos,z)}$ on the noisy test set of VoiceBank-DEMAND. The figure illustrates a general trend of increasing correlation coefficient with the number of iterations for most objective metrics. Notably, our $\text{VQScore}_{(cos,z)}$ exhibited exceptionally high correlations with BAK, OVR, and PESQ.

## B    COMPARISON OF DIFFERENT VQSCORES

Table 6: Linear correlation coefficient between **real** quality scores and various VQScores on different test sets.

|  | $\text{VQScore}_{(L_2,x)}$ | $\text{VQScore}_{(L_2,z)}$ | $\text{VQScore}_{(cos,x)}$ | $\text{VQScore}_{(cos,z)}$ |
|---|---|---|---|---|
| Tencent_wR | -0.0081 | -0.3709 | 0.0988 | **0.5865** |
| Tencent_woR | 0.4925 | -0.5983 | 0.5636 | **0.7159** |
| IUB_cosine | 0.0320 | -0.4266 | 0.1819 | **0.4880** |
| IUB_voices | 0.1764 | -0.8436 | 0.6943 | **0.8604** |

In Section 2.4, we discussed four combinations of distance metrics and targets for calculating VQScore. Table 6 presents the correlation coefficients between real quality scores and various VQScores on different test sets. It is worth noting that VQScores using $L_2$ as the distance metric are expected to exhibit a negative correlation with true quality scores (i.e., a larger distance implies poorer speech quality). The results demonstrate that employing cosine similarity in the code space ($z$) can significantly outperform the other alternatives.

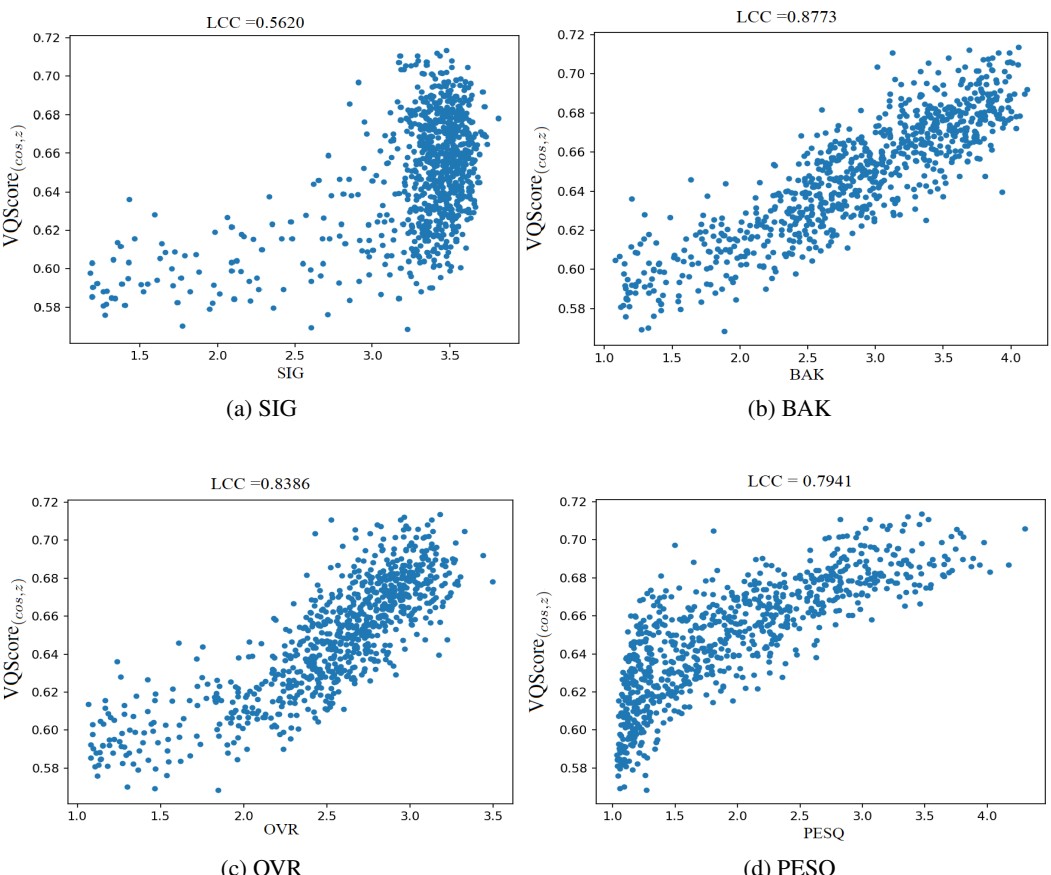

Figure 3: Scatter plots between various objective metrics and the proposed VQScore$_{(cos,z)}$ on the VoiceBank-DEMAND noisy test set. (a) SIG, (b) BAK, (c) OVR, and (d) PESQ.

## C SCATTER PLOTS FOR SPEECH QUALITY ESTIMATION

Figure 3 illustrates the scatter plots between the proposed VQScore$_{(cos,z)}$ and various objective metrics on the noisy test set of VoiceBank-DEMAND. From Figure 3 (a), it can be observed that the correlation between VQScore$_{(cos,z)}$ and SIG is low, particularly when the value of SIG is high. On the other hand, Figure 3 (d) reveals a low correlation between VQScore$_{(cos,z)}$ and PESQ when the value of PESQ is low. These findings suggest that modeling quality scores in extreme cases may present greater challenges. Furthermore, Figure 4 displays the scatter plots between the proposed VQScore$_{(cos,z)}$ and real subjective quality scores. Similar trends can be found in Figure 4 (c) and (d), indicating a low correlation between VQScore$_{(cos,z)}$ and real scores when the speech quality is poor.

## D FRAME-LEVEL SNR ESTIMATOR

Most machine-learning-based quality estimators are black-box, so people find it hard to understand the reason for their evaluation. On the other hand, from the definition of VQScore (Eq. 4), we can observe that the utterance score is based on summing up all the similarity scores of every frame. We, therefore, want to further verify that the proposed method can **localize** the frames where quality degrades (i.e., due to noise or speech distortion, etc.) in an utterance. Because most off-the-shelf metrics cannot provide such scores on a frame basis, here we use frame-level SNR as the ground truth. Given a synthetic noisy utterance, the frame-level SNR is calculated on the magnitude spectrogram for each frame. Since our preliminary experiments suggest that calculating the $L_2$ distance in the signal space has a higher correlation with SNR, here we define the predicted frame-based quality as:

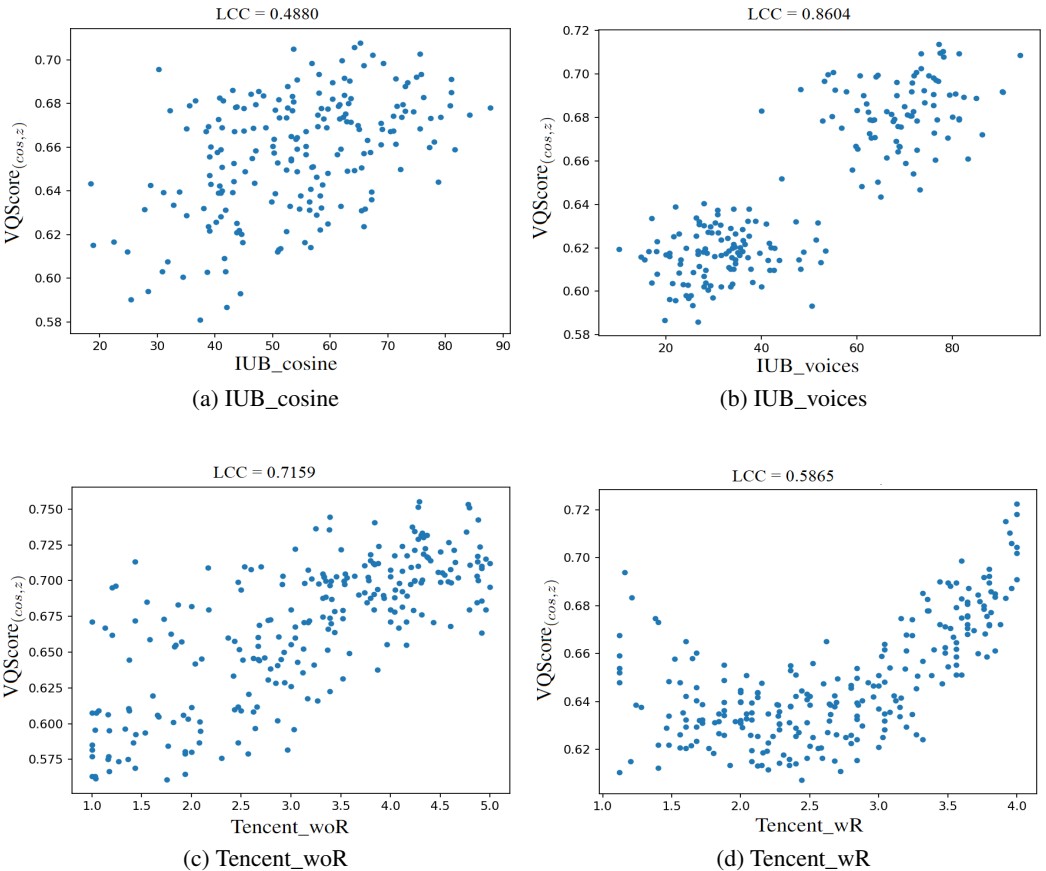

Figure 4: Scatter plots between real subjective quality scores and the proposed VQScore$_{(cos,z)}$ on (a) IUB_cosine, (b) IUB_voices, (c) Tencent_woR, and (d) Tencent_wR.

Table 7: Average linear correlation coefficient between frame-level SNR and the proposed method on the VoiceBank-DEMAND noisy test set.

|  | Supervised | Self-Supervised |
|---|---|---|
|  | Li et al. (2020) | Proposed |
| Frame-level SNR | 0.721 | **0.789** |

$$\frac{||X_t||_2}{||X_t - \hat{X}_t||_2} \tag{8}$$

The denominator is used to measure the reconstruction error and the numerator is for normalization.

In this experiment, we train the VQ-VAE with $L_2$ loss and evaluate the average correlation coefficient with ground-truth frame-level SNR on the VoiceBank-DEMAND noisy test set. Table 7 shows that our proposed metric can achieve a higher correlation than the supervised baseline (Li et al., 2020), which uses frame-level SNR as the model's training target. Figure 5 shows examples from the VoiceBank-DEMAND noisy test set, featuring spectrograms alongside their corresponding frame-level SNR and predicted frame-level quality using Eq. (8). Comparing Figure 5 (c) with (e), and (d) with (f), it can be observed that the overall shapes are similar, indicating a high correlation between them.

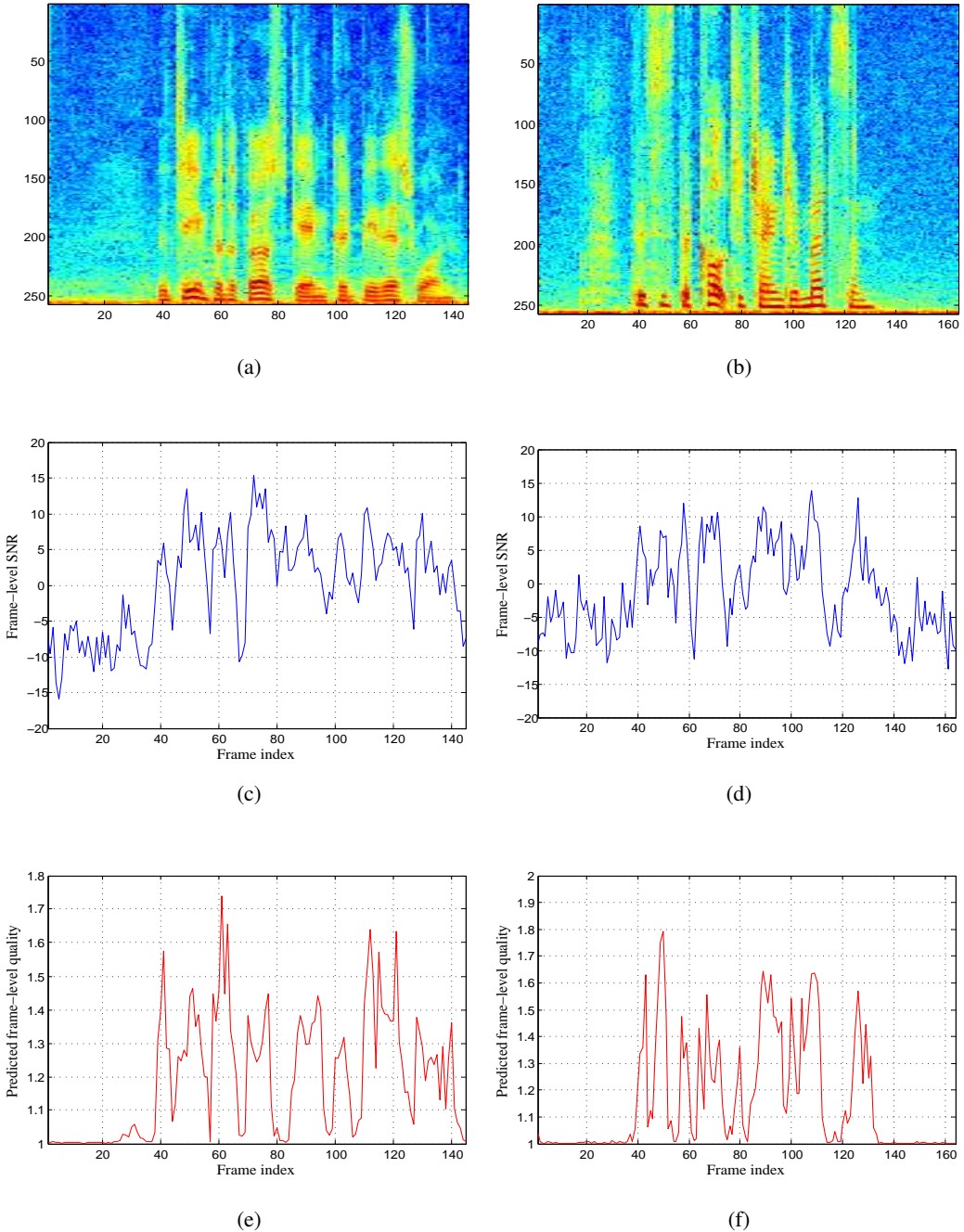

Figure 5: Examples of spectrogram, its corresponding frame-level SNR and the predicted frame-level quality. (c) and (d) are the frame-level SNR. (e) and (f) are our predicted frame-level quality.

# E   SPEECH ENHANCEMENT RESULTS ON THE DNS3 TEST SET

Table 8 displays the speech enhancement results on the DNS3 test set. Similar to the findings in DNS1 test set, it can be observed that applying AT in our model training can also further improve the scores of all the evaluation metrics. In addition, Proposed + AT can outperform Demucs, especially in the more mismatched conditions (i.e., Real or non-English cases).

Table 8: Comparison of different SE models on the DNS3 test set. Training data comes from the training set of VoiceBank-DEMAND.

| Subset | Model | Training data | SIG | BAK | OVRL |
|---|---|---|---|---|---|
| Real English | Noisy | - | **3.094** | 2.178 | 2.078 |
| | CNN-Transformer | (noisy, clean) pairs | 2.887 | 3.421 | **2.468** |
| | Demucs (Defossez et al., 2020) | (noisy, clean) pairs | 2.749 | 3.316 | 2.325 |
| | Wiener (Loizou, 2013) | - | 3.057 | 2.361 | 2.125 |
| | Proposed | clean speech | 2.844 | 3.157 | 2.305 |
| | Proposed + AT | clean speech | 2.888 | **3.468** | 2.456 |
| Real non-English | Noisy | - | 3.154 | 3.000 | 2.487 |
| | CNN-Transformer | (noisy, clean) pairs | **3.183** | 3.644 | 2.798 |
| | Demucs (Defossez et al., 2020) | (noisy, clean) pairs | 2.842 | 3.442 | 2.451 |
| | Wiener (Loizou, 2013) | - | 3.142 | 3.099 | 2.489 |
| | Proposed | clean speech | 3.120 | 3.602 | 2.733 |
| | Proposed + AT | clean speech | 3.179 | **3.726** | **2.820** |
| Synthetic non-English | Noisy | - | 3.165 | 2.597 | 2.300 |
| | CNN-Transformer | (noisy, clean) pairs | 3.053 | 3.590 | **2.645** |
| | Demucs (Defossez et al., 2020) | (noisy, clean) pairs | 2.716 | 3.526 | 2.374 |
| | Wiener (Loizou, 2013) | - | **3.174** | 2.749 | 2.361 |
| | Proposed | clean speech | 2.962 | 3.334 | 2.470 |
| | Proposed + AT | clean speech | 3.019 | **3.644** | 2.627 |
| Synthetic English | Noisy | - | **3.501** | 2.900 | 2.646 |
| | CNN-Transformer | (noisy, clean) pairs | 3.470 | **4.069** | **3.214** |
| | Demucs (Defossez et al., 2020) | (noisy, clean) pairs | 3.357 | 3.929 | 3.053 |
| | Wiener (Loizou, 2013) | - | 3.411 | 3.216 | 2.736 |
| | Proposed | clean speech | 3.365 | 3.937 | 3.062 |
| | Proposed + AT | clean speech | 3.381 | 4.039 | 3.117 |

## F  SPECTROGRAM COMPARISON OF ENHANCED SPEECH

Figure 6 and 7 present examples of enhanced spectrograms obtained from various SE models in the Real and Reverb conditions from the DNS1 test set, respectively. These figures visually reveal that the proposed self-supervised SE model exhibits good noise removal capabilities compared to other baselines.

## G  ADVERSARIAL TRAINING'S LEARNING CURVES

Figure 8 shows the adversarial training's learning curves on the VoiceBank-DEMAND noisy test set. From the curves, we can observe that the process of AT is quite stable. The scores of most evaluation metrics (except for SIG) first gradually increase and then converge to a better optimum compared to the initial (the result after Step 1 in Section 2.5). Compared to normal training, our AT only needs another forward and backward pass of the computational graph (i.e., adversarial attack, Eq. 5), therefore, the computation cost is roughly twice of normal training. However, as illustrated in the learning curves, AT is efficient and can converges quickly.

## H  DISTRIBUTION OF VQSCORE

To study the distribution of the VQScore, we first divide the test set into 3 subsets with equal size based on the sorted MOS. The first and the third subset corresponds to the group of the lowest and highest MOS, respectively. Figure 9 shows the histogram of the VQScore on the IUB test set, where blue and orange represent the first and the third set, respectively. Although there is some overlap in between for IUB_cosine, speech with higher MOS usually also have higher VQScore.

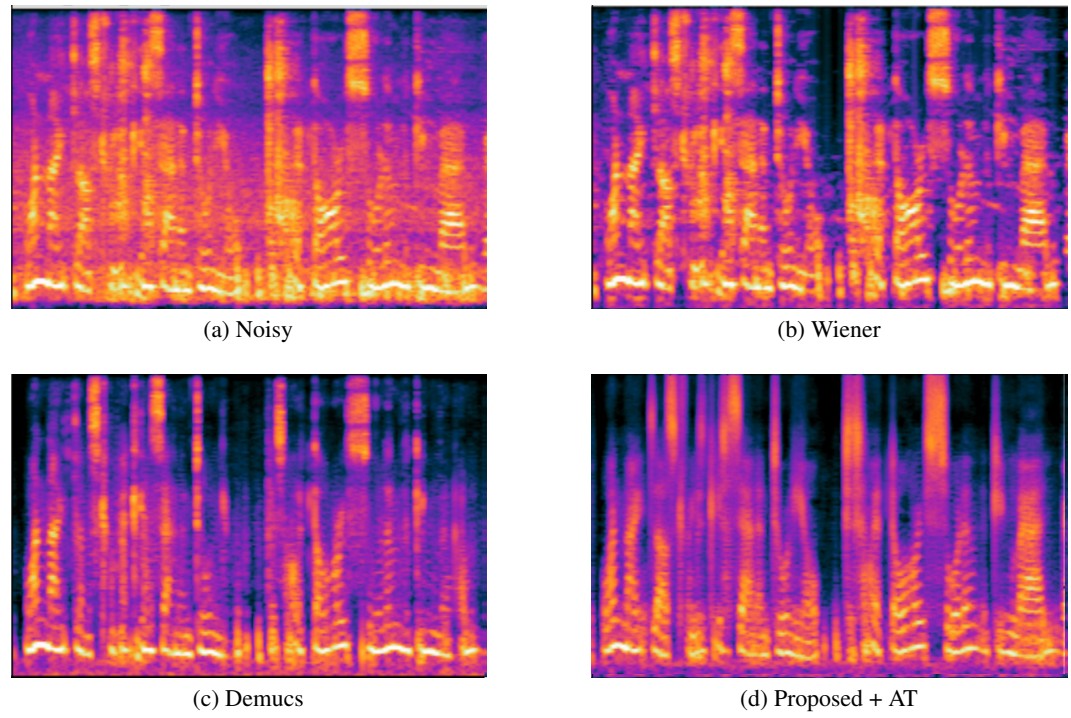

Figure 6: Spectrograms generated by different SE models. This utterance (realrec_fileid_10) is selected from the DNS1 **Real** test set. (a) Noisy, (b) Wiener, (c) Demucs, and (d) Proposed + AT.

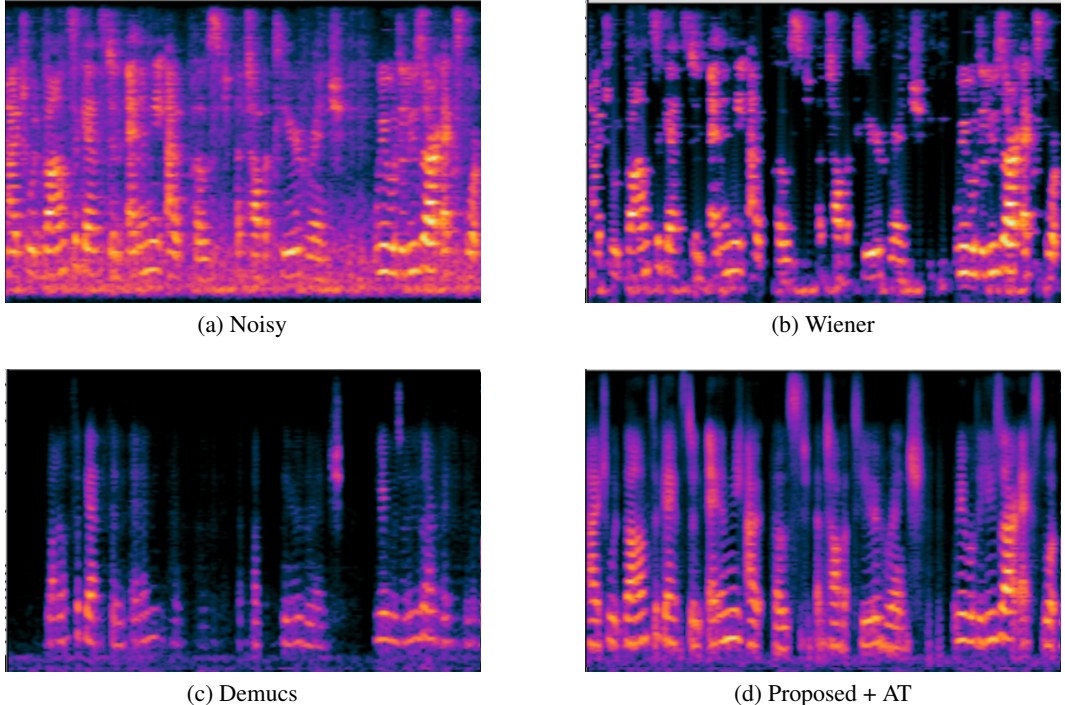

Figure 7: Spectrograms generated by different SE models. This utterance (reverb_fileid_5) is selected from the DNS1 **Reverb** test set. (a) Noisy, (b) Wiener, (c) Demucs, and (d) Proposed + AT.

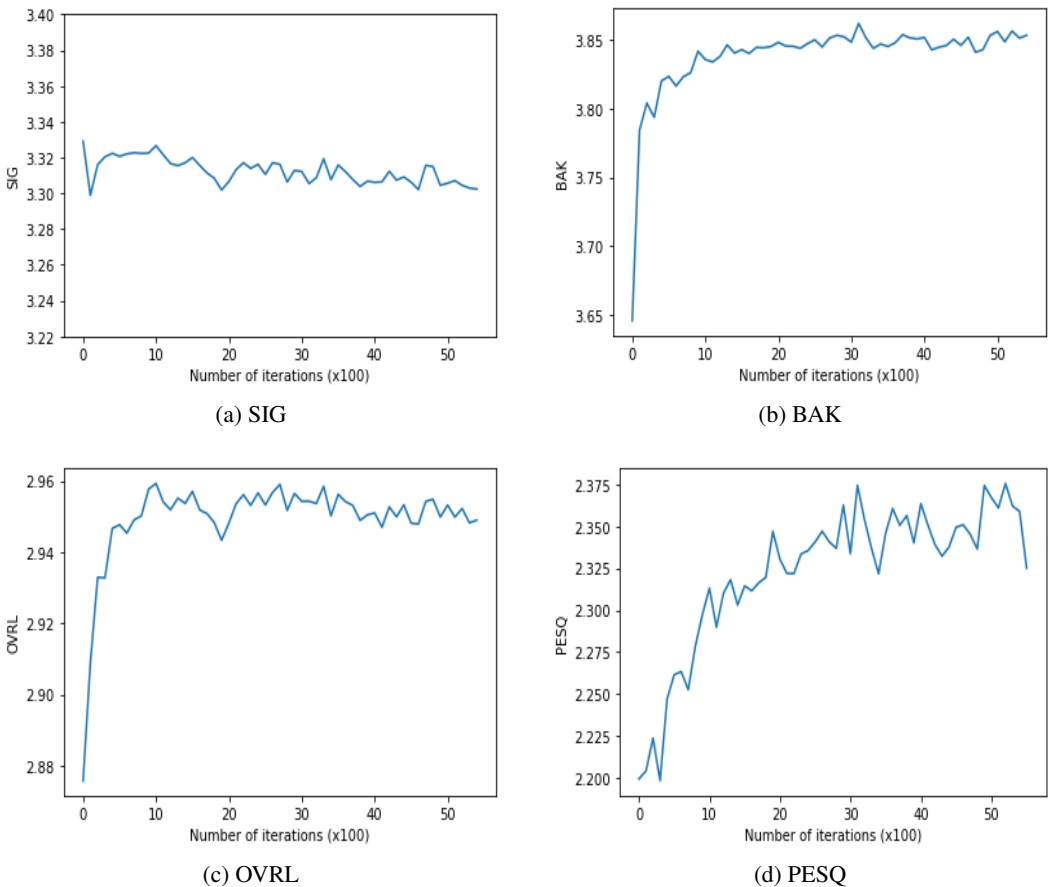

(a) SIG

(b) BAK

(c) OVRL

(d) PESQ

Figure 8: Adversarial training's learning curves on the VoiceBank-DEMAND noisy test set. The starting point is the result after Step 1 in Section 2.5 (i.e., VQ-VAE trained on clean speech has converged). (a) SIG, (b) BAK, (c) OVRL, and (d) PESQ.

## I SENSITIVITY TO HYPER-PARAMETERS

In this section, we study the effect of hyper-parameters on model performance. All the hyper-parameters were decided based on the performance of DNSMOS (OVRL) on the validation set. For quality estimation, it is the LCC between DNSMOS (OVRL) and VQScore. For the speech enhancement, it is the score itself. We first investigate the influence of codebook size on quality estimation. From Table 9, we can observe that except for very small codebook dimensions (i.e., 16), the performance is quite robust to codebook number and dimension. The case for speech enhancement is similar, except for very small codebook dimensions and numbers, the performance is robust to the codebook setting. We next investigate the effect of setting different $\beta$ in Eq. 3. Table 10 shows that the SE performance is also robust to different $\beta$ which aligns with the observation made in (Van Den Oord et al., 2017).

Table 9: LCC between DNSMOS (OVRL) and VQScores under different codebook sizes.

| Codebook size (number, dim) | LCC |
|---|---|
| (1024, 32) | 0.8332 |
| (2048, 16) | 0.7668 |
| **(2048, 32)** | **0.8386** |
| (2048, 64) | 0.8317 |
| (4096, 32) | 0.8297 |

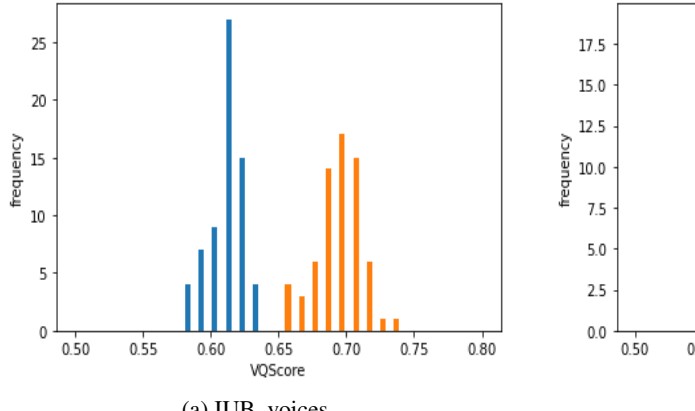 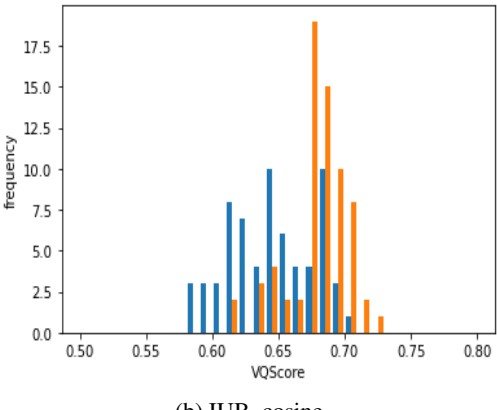

(a) IUB_voices                                (b) IUB_cosine

Figure 9: Histogram of VQScore. Blue and orange represent the sets with lower and higher MOS scores, respectively. (a) IUB_voices, (b) IUB_cosine.

Table 10: DNSMOS (OVRL) of enhanced speech from our models trained with different $\beta$.

| $\beta$ | DNSMOS (OVRL) |
|---|---|
| 1 | 2.865 |
| 2 | 2.872 |
| **3** | **2.876** |

## J    MODEL COMPLEXITY

In this section, we compare model complexity based on the number of parameters and the number of computational operations as shown in Table 11. MACs stand for multiply–accumulate operation and are calculated based on 1 sec of audio input. Because NyTT doesn't release the model, it is difficult to accurately estimate its model complexity. However, its model structure is based on CNN-BLSTM, so we can expect it to have higher model complexity compared to MetricGAN-U, which is based on simple BLSTM. CNN-Transformer is the supervised version (and removing VQ) of our proposed model and hence has a similar model complexity. Demucs is a CNN encoder and decoder framework with BLSTM in between to model temporal relationships. Because directly models the waveform, its model complexity is significantly higher than others.

Table 11: Model complexity for the proposed approach and baselines.

|  | Params (M) | MACs (G) |
|---|---|---|
| CNN-Transformer | 2.51 | 0.32 |
| Demucs | 60.81 | 75.56 |
| MetricGAN-U | 1.90 | 0.24 |
| NyTT | - | - |
| Proposed | 2.51 | 0.32 |

## K    STATISTICAL SIGNIFICANCE

In Table 12, we report the T-test of DNSMOS (OVR) between Proposed + AT and different baselines on the DNS1 test set to show the statistical significance. In the table, the results shown in bold represent Proposed + AT is statistically significant (p-value<0.05) better than the baseline. It can be observed that Proposed + AT is significantly better than most of the baselines (noisy, Wiener, and Demucs) and is comparable to CNN-Transformer.

Table 12: P-value of DNSMOS (OVR) between Proposed + AT and baselines on the DNS1 test set.

| | Noisy | Wiener | Demucs | CNN-Transformer |
|---|---|---|---|---|
| Real | **1.35e-36** | **4.84e-31** | **8.97e-07** | **0.0001** |
| Noreverb | **7.84e-43** | **1.18e-47** | **0.0042** | 0.141 |
| Reverb | **9.20e-54** | **8.70e-34** | **3.21e-08** | 0.205 |

## L   ASR RESULTS OF ENHANCED SPEECH

In this section, we apply Whisper-medium (Radford et al., 2023) as the ASR model and compute the word error rate (WER) of speech generated by different SE models (with dry/wet knob technique as proposed in (Defossez et al., 2020)) on the VoiceBank-DEMAND noisy test set. Table 13 shows that all the SE can improve the WER performance, and our proposed method can achieve the lowest WER.

Table 13: WER of speech generated by different SE models on the VoiceBank-DEMAND noisy test set.

| | Noisy | Wiener | Demucs | CNN-Transformer | Proposed |
|---|---|---|---|---|---|
| Whisper ASR | 14.25 | 12.60 | 13.75 | 11.84 | **11.65** |

## M   LIMITATIONS AND FUTURE WORKS

1) **Speech quality estimation**:

Our preliminary experiment results show that the VQScore can obtain LCC around 0.46 with the VoiceMOS 2022 challenge test set (Huang et al., 2022). This result is comparable to SpeechLMScore (Pre) (0.452) but worse than SpeechLMScore(LSTM)+rep (0.582). One possible reason is that VQScore trained on LibriSpeech clean-460 hours only uses <10% training data of SpeechLMScore. Another possible reason is that if we observe each frame generated by a TTS system, it resembles a clean frame. In the TTS evaluation, people may focus more on global conditions such as **naturalness**, etc. In other words, it cares more about the relation of each frame with each other. On the other hand, VQScore pays more attention to the degradation of each frame.

As can be observed in Eq. 4, the VQScore is based on the average of the cosine similarity of input frames. However, people may put larger weights on the frames with louder volume when evaluating speech quality. It is difficult to design/learn the weights of each frame in the **unsupervised** setting. If we extend to a **semi-supervised** framework, we believe this consideration will bring further improvements.

2) **Speech enhancement**:

As discussed in the previous section, we observed a more pronounced speech distortion in the speech generated by our method. In fact, the results of our listening test indicate that while our model receives higher scores for noise removal (BAK), its speech distortion score (SIG) is comparatively worse than that of conventional methods. Further analysis revealed that the primary source of speech distortion may come from the **finite** combination of the discrete tokens in the VQ module. In summary, while the VQ module contributes to the model's great noise removal capability, it simultaneously introduces speech distortion. One possible solution is to fuse the distorted enhanced speech with the original noisy speech to recover some over-suppressed information (Hu et al., 2023). Our future efforts in the development of this SE approach will be dedicated to mitigating the speech distortion caused by the VQ module.

