# OpenReview forum: "Self-Supervised Speech Quality Estimation and Enhancement Using Only Clean Speech"
_ICLR.cc/2024/Conference — ICLR 2024 poster_

### Official Review · Reviewer_KXQH · 2023-10-29

**Soundness:** 4 excellent
**Presentation:** 3 good
**Contribution:** 3 good
**Rating:** 8
**Confidence:** 5

**Summary:**

Authors propose to use VQ-VAE in self-supervised audio quality estimation and enhancement based on solely training with clean audio. Idea is to correlate quantization error in the latent space to quality metrics. Speech enhancement is then performed by the way of finetuning using the adversarial noise. So still no need to feed in noisy samples.

**Strengths:**

Key idea of using the quantization error in VQ-VAE as the qualityt metric is novel as far as I know and also the idea is quite neat. I like it a lot. Enhacement idea based on this innovation is also quite nice. Experimental results do support the hypotheses.

**Weaknesses:**

- Very little theoretical analysis is found in the paper. When would the proposed method work and it would fail? Can anything be said about it?
- Key parameters are not empirically, nor theorerically assessed. Especially codebook size appears to be extremely critical parameter.
- Significance testing should be reported for each computed correlation.

**Questions:**

- How was the commitment weight \beta = 3 decided?
- Basically quantization error is the measure that you are using and it for sure does make sense. It would be interesting to see whether some distrubutional arguments can be made about the quantization errors. Note that those errors are scalar quantities and thus could be easily plotted and visually inspected.

---

> ### Author Response · Authors · 2023-11-19
> **Response to Reviewer KXQH (Q1~Q3)**
>
> We appreciate Reviewer KXQH likes our idea of VQScore and considering our work is with novelty, and your comments are very helpful for us on improving presentation quality and clarifications with wider impacts. Please find the corresponding responses below:
>
> **Q1: Very little theoretical analysis is found in the paper. When would the proposed method work and it would fail? Can anything be said about it?**
>
> - Our preliminary experiment results show that the VQScore is only comparable to SpeechLMScore in the VoiceMOS 2022 challenge test set (a challenge to predict MOS for synthesized speech from TTS and voice conversion). One possible reason is that VQScore trained on LibriSpeech clean-460 hours only uses <10% training data of SpeechLMScore. Another possible reason is that if we look at each frame generated by a TTS system, it may look like a clean frame. However, in the TTS evaluation, the evaluation may focus more on global conditions such as **NATURALNESS**, etc. In other words, it cares more about the relation of each frame with the other. On the other hand, VQScore pays more attention to the degradation of each frame.
>
> **Q2: Key parameters are not empirically, nor theorerically assessed. Especially codebook size appears to be extremely critical parameter**
>
> - Thank you for pointing this out, we have added more description in the paper (section I in the Appendix) to describe how we decide the hyper-parameters. We decided the hyper-parameters (e.g., \beta, and codebook size, etc.) based on the performance of DNSMOS (OVRL) on validation set. For quality estimation, it is the linear correlation coefficient (LCC) with VQScore. For the speech enhancement, it is the score itself.
>
> - LCC between DNSMOS (OVRL) and VQScores with different codebook sizes are shown in Table r1. From this table, we can observe that except for very small codebook dimensions (i.e., 16), the performance is quite robust to codebook number and dimension. The case for speech enhancement is similar, except for very small codebook dimensions and numbers, the performance is robust to the codebook setting. These results have been added to the Appendix of the paper.
>
> - Table r1: LCC between DNSMOS (OVRL) and VQScores with different codebook size.
> | Codebook size (number, dim)     | LCC |
> | :----:        |    :----:   |
> | (1024, 32)| 0.8332|
> | (2048, 16)| 0.7668 |
> | **(2048, 32)**|	**0.8386** |
> | (2048, 64)|	0.8317 |
> | (4096, 32)|	0.8297 |
>
>
>
> **Q3: Significance testing should be reported for each computed correlation.**
>
> - Thank you for the suggestion. In the following tables, we report the T-test between Proposed + AT and different baselines on the DNS1 test set to show the statistical significance. **!!Please check Section K in the Appendix for the colored version.!!** In the tables, results shown in red, and blue represent Proposed + AT is statistically significant (p-value<0.05) better and worse than the baseline, respectively (results with black color represent no statistically significant). From the tables, we can observe that Proposed + AT is usually statistically significant better on the DNSMOS (BAK) and DNSMOS (OVR) which implies better noise removal ability. This improvement mainly comes from VQ and AT.
>
> - Table r2: P-value of DNSMOS (SIG) between Proposed + AT and baselines
> on the DNS1 test set.
> || Noisy     | Wiener | Demucs| CNN-Transformer|
> | :----:  |    :----:   |   :---  |    :----:   |   :----:   |
> | Real  |   0.683   |   0.128 |   0.0191   |   0.0025  |
> | Noreverb	|   0.743   |   2.29e-14 |   0.00126   |   0.265  |
> | Reverb	|   4.35e-24   |   3.46e-09 |   1.55e-11   |   0.0019  |
>
>
> - Table r3: P-value of DNSMOS (BAK) between Proposed + AT and baselines
> on the DNS1 test set.
> || Noisy     | Wiener | Demucs| CNN-Transformer|
> | :----:  |    :----:   |   :---  |    :----:   |   :----:   |
> | Real  |  3.77e-104	 | 3.90e-83	 | 3.53e-14	 | 3.01e-10  |
> | Noreverb	|   3.79e-61| 4.69e-49| 2.97e-08| 0.0018  |
> | Reverb	|   4.74e-91| 1.92e-46| 0.114| 0.422 |
>
>
> - Table r4: P-value of DNSMOS (OVR) between Proposed + AT and baselines
> on the DNS1 test set.
> || Noisy     | Wiener | Demucs| CNN-Transformer|
> | :----:  |    :----:   |   :---  |    :----:   |   :----:   |
> | Real  |  1.35e-36  | 	4.84e-31 | 8.97e-07	 | 0.0001  |
> | Noreverb	|   7.84e-43| 1.18e-47| 0.0042| 0.141  |
> | Reverb	|   9.20e-54| 8.70e-34| 3.21e-08| 0.205 |

---

> > ### Author Response · Authors · 2023-11-19
> > **Response to Reviewer KXQH (Q4~Q5)**
> >
> > **Q4: How was the commitment weight \beta = 3 decided?**
> >
> > - As pointed out in the original paper of VQ-VAE [r1]: “We found the resulting algorithm to be quite robust to β, as the results did not vary for values of β ranging from…”. Our ablation study shown in the following table also appears a similar trend. Although, the performance is not sensitive to β, we still select the β based on the performance on the validation set. Specifically, the DNSMOS (OVRL) of the validation set.
> >
> > - Table r2: DNSMOS (OVRL) of enhanced speech from models trained with different β.
> > | β      | DNSMOS (OVRL) |
> > | :----:        |    :----:   |
> > | 1      | 2.865       |
> > | 2	  | 2.872 |
> > | **3**	  | **2.876**      |
> >
> >
> > [r1] A. Van Den Oord, et al., “Neural discrete representation learning,” Nips, 2017.
> >
> > **Q5: Basically quantization error is the measure that you are using and it for sure does make sense. It would be interesting to see whether some distrubutional arguments can be made about the quantization errors. Note that those errors are scalar quantities and thus could be easily plotted and visually inspected.**
> >
> > - Thank you for this suggestion, it is a very interesting direction to observe the distribution!! We have added the distribution of the VQScore on the IUB test set in the updated appendix (section H). To study the distribution of the VQScore, we first divide the test set into 3 subsets with equal size based on the sorted MOS. The first and the third subset corresponds to the group of the lowest and highest MOS, respectively. Figure 9 shows the histogram of the VQScore on the IUB test set, where blue and orange represent the first and third sets, respectively. Although there is some overlap in between for IUB_cosine, speech with higher MOS usually also has higher VQScore.

---

> > > ### Author Response · Authors · 2023-11-22
> > > **Waiting for the reply**
> > >
> > > Dear Reviewer KXQH,
> > >
> > > In response to the feedback, we've conducted additional experiments to enhance the depth and robustness of our work. This includes:
> > >
> > > 1) How we decide the hyper-parameters.
> > >
> > > 2) Significance testing of the experimental results.
> > >
> > > 3) Distribution of quantization error.
> > >
> > > We believe our detailed responses provide clarity on the concerns raised and are happy to have further discussion. Please let us know if you have any additional concerns before the **approaching** deadline of the Author-Reviewer discussion phase. Your feedback is pivotal to the quality of our work, and we earnestly await your thoughts.
> > >
> > > Thank you for your time and understanding.

---

> > > ### Comment · Reviewer_KXQH · 2023-11-22
> > > **Rebuttal looks good to me**
> > >
> > > In terms of my concerns, I find that authors responded clearly and insightfully. The only concern left for me is Q1, I feel that authors  did not try to really answer my, admittedly, fairly vague comment.
> > >
> > > However, I am willing to raise my grade by one point.

---

> > > > ### Author Response · Authors · 2023-11-23
> > > > **Response to Reviewer KXQH**
> > > >
> > > > Thank you very much for your positive feedback! Due to the limited time remaining for the rebuttal, we will provide more discussion of the limitations of our model in the revised manuscript.

---

### Official Review · Reviewer_eoAZ · 2023-10-31

**Soundness:** 3 good
**Presentation:** 3 good
**Contribution:** 3 good
**Rating:** 6
**Confidence:** 4

**Summary:**

Contributions of the paper are two-fold; first, the authors propose a speech quality measure based on the comparison of speech embeddings before and after vector quantization using a VQ-VAE. Two metrics were used for comparison: a L_2 norm and a cosine similarity metric. During the experimental phase, the authors compare the proposed metric with some previously proposed objective speech quality metrics on four data sets that contain human perception metrics.  Among the metrics used for comparison, they included SNR, PESQ, SIG, BAK, and OVR.  Based on the SNR results, the authors also suggest that the proposed method can estimate SNR in a frame-based approach. Second,  the paper presents a model distillation approach using a two steps learning process where a noise component is learned such that it minimizes the performance of the quantization process, and a second step where the encoder of the student model is trained to revert that behavior, making it more robust to noisy samples. The decoder is trained to reduce the reconstruction error, as in any denoising approach.

**Strengths:**

The paper addresses a problem of interest in the state of the art that does not have a clear solution. The proposed solution for speech quality assessment is simple, yet it could be effective. The proposed method for speech enhancement requires only clean data and the proposed adversarial training is an interesting alternative to

**Weaknesses:**

The novelty of the paper is limited. Quality metrics comparing embedding has already been proposed in multimodal or generative contexts including speech such as the Fréchet Audio Distance. Moreover, the authors found a low correlation between the proposed score and the quality benchmarks when the speech quality is poor, limiting the proposed measure's reliability. The results of the proposed approach for speech enhancement are still behind those of supervised models.

**Questions:**

- Notation of equations 5 and 6 is inconsistent. According to Eq. 5, Lce is a function of two arguments, but Eq. 6 does not develop it correctly. Notation in general, should be reviewed.
- The authors should show evidence of the training stability during the proposed adversarial training. ¿Is there any risk that during training, the model collapses to select the same token?
- Other adversarial training strategies suffer from the high cost of generating adversarial samples, and the proposed approach does not seem to do differently. The authors should include analyses regarding computational load and scalability.
- The paper should also include experiments to support the claims that the proposed approach can exhibit better generalization capabilities to new domains than supervised models. Telephony speech or artificially generated speech should also be included.

---

> ### Author Response · Authors · 2023-11-19
> **Response to Reviewer eoAZ (Q1~Q3)**
>
> We sincerely appreciate Reviewer eoAZ for considering our work addresses a problem of interest that does not have a clear solution. Your suggestion and comments for further enhancing the submission quality are very helpful. Please find the responses below:
>
> **Q1: The novelty of the paper is limited. Quality metrics comparing embedding has already been proposed in multimodal or generative contexts including speech such as the Fréchet Audio Distance.**
>
> - Thank you for the comment. One main difference between our VQscore and Fréchet Audio Distance (FAD) is that FAD cannot evaluate the quality of each **INDIVIDUAL** audio clip as VQscore. As pointed out in the paper of FAD [r1] (section 3.1 Definition): “Unlike existing audio evaluation metrics, FAD does not look at individual audio clips, but instead compares embedding statistics generated on the whole evaluation set with embedding statistics generated on a large set of clean music (e.g. the training set).” In other words, FAD compares the distance between two sets (distributions), and the number of elements in the set has to be large enough to make FAD reliable. **In summary, FAD can only be used to evaluate the performance of a generative model, not a single audio clip.**
>
> - As far as we know, we are the **FIRST** one to propose using quantization error to measure speech quality without the need for any quality label during training. **Note that being able to calculate the quantization error is a unique property of the VQ-VAE that other autoencoders do not have.**
>
> [r1] Kilgour, K., Zuluaga, M., Roblek, D., & Sharifi, M. (2018). Fr\'echet Audio Distance: A Metric for Evaluating Music Enhancement Algorithms. arXiv preprint arXiv:1812.08466.
>
> **Q2: Moreover, the authors found a low correlation between the proposed score and the quality benchmarks when the speech quality is poor, limiting the proposed measure's reliability. The results of the proposed approach for speech enhancement are still behind those of supervised models.**
>
> - It is difficult to ask an objective quality metric to always align with subjective scores perfectly in **EVERY** condition. **For example, in Fig. 2 of [r2], the widely-used quality metrics, PESQ and POLQA also perform less reliably when the speech quality is poor.** From our experimental results shown in Table 2, our VQScore is comparable to supervised SOTA (DNSMOS), which is trained on large-scale paired data while our VQscore doesn’t need any label for model training.
>
> - Our proposed SE model is only behind supervised models in the **matched** condition (Table 3), where the training and testing sets were from the **SAME** source. However, in practical applications, where the training and testing sets were from **DIFFERENT** sources (Tables 4 and 5), our self-supervised SE model shows better generalization capabilities than the supervised Demucs model (from Meta) and is comparable to CNN-Transformer, which has a similar model architecture as our self-supervised SE.
>
> - We have added an ASR experimental result in Table r1. We apply Whisper [r3] as the ASR model and compute the WER of speech generated by different SE models [r4] on the VoiceBank-DEMAND noisy test set. The results show that all the SE can improve the WER performance, and our proposed method can achieve the **lowest** WER.
>
> - Table r1: WER of speech generated by different SE models on the
> VoiceBank-DEMAND noisy test set.
> || Noisy|	Wiener|	Demcus|	CNN-Transformer|	Proposed |
> | :----:        |    :----:   |    :----:   |  :----:        |    :----:   |    :----:   |
> |Whisper ASR|	14.25|	12.60|	13.75|	11.84|	**11.65**|
>
>
> [r2] Reddy, C. K., Gopal, V., & Cutler, R. (2021, June). DNSMOS: A non-intrusive perceptual objective speech quality metric to evaluate noise suppressors. IEEE International Conference on Acoustics, Speech and Signal Processing (ICASSP), 2021.
>
> [r3] Radford, A., Kim, J. W., Xu, T., Brockman, G., McLeavey, C., & Sutskever, I. (2023, July). Robust speech recognition via large-scale weak supervision. In International Conference on Machine Learning (pp. 28492-28518). PMLR.
>
> [r4] Defossez, A., Synnaeve, G., & Adi, Y. (2020). Real time speech enhancement in the waveform domain. arXiv preprint arXiv:2006.12847.
>
> **Q3: Notation of equations 5 and 6 is inconsistent. According to Eq. 5, Lce is a function of two arguments, but Eq. 6 does not develop it correctly. Notation in general, should be reviewed.**
>
> - Sorry for the confusion. The additional argument Cv in Eq. 6 is the **FIXED** dictionary which can be treated as constant during AT. We have modified Eq.5 in the paper to include Cv as a function input to avoid confusion.

---

> > ### Author Response · Authors · 2023-11-19
> > **Response to Reviewer eoAZ (Q4~Q6)**
> >
> > **Q4 :The authors should show evidence of the training stability during the proposed adversarial training. ¿Is there any risk that during training, the model collapses to select the same token?**
> >
> > - Different from GAN, which makes two models (generator and discriminator) compete with each other, our proposed AT **doesn’t contain a discriminator** (please see eqs (5)-(7)), therefore, the AT is pretty stable.
> >
> > - We have added learning curves of AT on the validation set in the Appendix (Section G) of the paper. The starting point (iteration 0) is the result after Step 1 in Section 2.5 (VQ-VAE trained on clean speech has converged). From the curves, we can observe that the process of AT is very stable. The scores of evaluation metrics (except for SIG) first gradually increase and then converge to a better optimum compared to the initial, which shows the effectiveness and stability of AT.
> >
> > - The model collapse usually happens in the case of a **UNCONDITIONAL GAN**, where the input is **pure noise**, and the model only generates certain classes of data (e.g., dog, and cat, etc.). In the conditional case (noisy speech as input in our case), because of the strong input prior, model collapse seldom happens.
> >
> > **Q5: Other adversarial training strategies suffer from the high cost of generating adversarial samples, and the proposed approach does not seem to do differently. The authors should include analyses regarding computational load and scalability.**
> >
> > - Unlike the iterative-fast gradient sign method (I-FGSM) [r5] which requires an iterative procedure to generate adversarial samples, the computational cost of our method is similar to FGSM [r6] which only needs a single pass to generate adversarial samples.
> > Compared to normal training (such as eq (7)), our adversarial training only needs another forward and backward pass of the computational graph (eq (5)), therefore, the computation cost is roughly **twice** of normal training. However, as illustrated in the learning curves on the validation set in the updated Appendix (Fig. 8), the AT training is efficient and can converge within **5000 iterations** (within **2 hours** using a common V100 GPU). Therefore, scalability should not be a concern.
> >
> > [r5] Kurakin, A., Goodfellow, I., & Bengio, S. (2016). Adversarial examples in the physical world.
> >
> > [r6] Goodfellow, I. J., Shlens, J., & Szegedy, C. (2014). Explaining and harnessing adversarial examples. arXiv preprint arXiv:1412.6572.
> >
> >
> > **Q6: The paper should also include experiments to support the claims that the proposed approach can exhibit better generalization capabilities to new domains than supervised models. Telephony speech or artificially generated speech should also be included.**
> >
> > - **About speech quality estimation:**
> >
> > - The test sets used for speech quality estimation actually come from many new domains as pointed out in the paper. For the VOiCES dataset, acoustic conditions such as foreground speech (reference), low-pass filtered reference (anchor), and reverberants were included. For the COSINE dataset, close-talking mic (reference) and chest or shoulder mic (noisy) data were provided. We even include the Tencent corpus, which covers many acoustic conditions encountered in an **online meeting in Chinese speech**.
> >
> > - **About speech enhancement:**
> >
> > - In fact, we have presented several experiment results (Tables 4, 5, and 8) to support our self-supervised SE model exhibits better generalization capabilities to **NEW** domains. Table 4 is based on the DNS1 test set, and contains three acoustic conditions, 1) without reverberation (noreverb), 2) noisy reverberant speech (reverb), and 3) noisy real recordings (real). Table 8 is based on the DNS3 test set, and it can be divided into subcategories based on realness (real or synthetic) and language (English or non-English). Therefore, we have verified the generalization capabilities in different conditions (**acoustic and language**, etc.)
> > Under these new domains, our self-supervised SE model shows better generalization capabilities than the supervised Demucs model.

---

> > > ### Author Response · Authors · 2023-11-22
> > > **Waiting for the reply**
> > >
> > > Dear Reviewer eoAZ,
> > >
> > > In response to the feedback, we've conducted additional experiments to enhance the depth and robustness of our work. This includes:
> > >
> > > 1) The difference between the proposed VQScore and Fréchet Audio Distance(FAD). Based on the comparison of the distance between two **sets (distributions)**, FAD can only be used to evaluate the performance of a **system**, not a single audio clip as our VQScore. As far as we know, we are the **FIRST** one to propose using quantization error to measure speech quality **without** the need for any quality label during training. **Note that being able to calculate the quantization error is a unique property of the VQ-VAE that other autoencoders do not have.**
> > >
> > > 2) An analysis of the computational load of AT, and show its efficiency and stability by providing learning curves.
> > >
> > > We believe our detailed responses provide clarity on the concerns raised and are happy to have further discussion. Please let us know if you have any additional concerns before the **approaching** deadline of the Author-Reviewer discussion phase. Your feedback is pivotal to the quality of our work, and we earnestly await your thoughts.
> > >
> > > Thank you for your time and understanding.

---

> ### Comment · Reviewer_eoAZ · 2023-11-22
>
> Thank you for your answer. You clarified many of my previous questions. Regarding the FDA metric, It is clear that FDA compares the distributions of two sets of audio embeddings because it was proposed in the context of the audio generation task, where there are no target outputs to compare with. Notwithstanding the context, the comparison of embeddings as a quality metric was proposed in there. Anyway, the proposal of comparing embedding and quantized embeddings in VQ-VAE is an interesting alternative for individual evaluations.
>
> The concerns about the model collapsing are not exclusive of GANs, as the authors claimed. The literature has reported problems of codebook collapse in VQ-VAE models, which is analogous to mode collapse in continuous generative models. So, concerns in this respect remain.

---

> ### Author Response · Authors · 2023-11-22
> **Response to Reviewer eoAZ**
>
> **We are happy to hear that some of your concerns are addressed.** Regarding codebook collapse in VQ-VAE, we have tried a technique proposed in SoundStream [r1] to increase codebook usage by replacing dead codes. However, using this technique only brings limited improvement. In fact, in our experience, **as long as the codebook size is large enough (see Table r1 in https://openreview.net/forum?id=ale56Ya59q&noteId=7IdAcnl0jE), codebook collapse is not a serious problem.**
>
> Please note that the purpose of this paper is **NOT** to solve the codebook collapse in VQ-VAE. We try to apply VQ-VAE to solve some difficult challenges with **novel** methods.
>
>
> [r1] Zeghidour, N., Luebs, A., Omran, A., Skoglund, J., & Tagliasacchi, M. (2021). Soundstream: An end-to-end neural audio codec. IEEE/ACM Transactions on Audio, Speech, and Language Processing, 30, 495-507.

---

### Official Review · Reviewer_DUwZ · 2023-11-02

**Soundness:** 3 good
**Presentation:** 3 good
**Contribution:** 3 good
**Rating:** 8
**Confidence:** 3

**Summary:**

In this paper authors propose VQScore to measure speech quality, which is based on the quantization error of VQ-VAE. It's a self-supervised metric without paired speech and noisy data in training. Based on it, the authors propose to improve speech enhancement with self-distillation with adversarial training. Experimental results show the effectiveness of the proposed methodology.

**Strengths:**

The proposed methodology is technical sound. Its training uses clean speech data only, and this helps reduce dependencies on noisy/clean speech pairs to develop models for speech quality measure and speech enhancement. Overall, this paper clearly describes the proposed approach, with well designed experiments and analysis.

**Weaknesses:**

I think the experimental section could be further strengthened with more details added. Please see the Questions section below.

**Questions:**

1. How to determine the values for several hyper-parameters, e.g. \beta in equation (3), codebook size etc.
2. For the results tables, could authors include std to show if difference is statistically significant?
3. For Table 3, could authors add a short summary about comparing model complexity for the proposed approach vs. baselines?

---

> ### Author Response · Authors · 2023-11-19
> **Response to Reviewer DUwZ (Q1~Q2)**
>
> We appreciate Reviewer DUwZ for considering our work is clear with well-designed analysis and very thank your reviewing efforts and advice. Your comments for improvement are professional and constructive. Please find the responses below:
>
> **Q 1:	How to determine the values for several hyper-parameters, e.g. \beta in equation (3), codebook size etc.**
>
> - Thank you for pointing this out, we have added more description in the paper (section I in the Appendix) to describe how we decide the hyper-parameters. We decided the hyper-parameters (e.g., \beta, and codebook size, etc.) based on the performance of DNSMOS (OVRL) on validation set. For quality estimation, it is the linear correlation coefficient (LCC) with VQScore. For the speech enhancement, it is the score itself.
>
> - As pointed out in the original paper of VQ-VAE [r1]: “We found the resulting algorithm to be quite robust to β, as the results did not vary for values of β ranging from…”. Our ablation study shown in the following table shows a similar trend. Although the performance is not sensitive to β, we still select β based on the performance on the validation set. Specifically, the DNSMOS (OVRL) of the validation set.
>
> - Table r1: DNSMOS (OVRL) of enhanced speech from models trained with different β.
> | β      | DNSMOS (OVRL) |
> | :----:        |    :----:   |
> | 1      | 2.865       |
> | 2	  | 2.872 |
> | **3**	  | **2.876**      |
>
>
> - LCC between DNSMOS (OVRL) and VQScores with different codebook sizes are shown in Table r2. From this table, we can observe that except for very small codebook dimensions (i.e., 16), the performance is quite robust to codebook number and dimension. The case for speech enhancement is similar, except for very small codebook dimensions and numbers, the performance is robust to codebook setting. These results have been added to the Appendix of the paper.
>
> - Table r2: LCC between DNSMOS (OVRL) and VQScores under different codebook size.
> | Codebook size (number, dim)     | LCC |
> | :----:        |    :----:   |
> | (1024, 32)| 0.8332|
> | (2048, 16)| 0.7668 |
> | **(2048, 32)**|	**0.8386** |
> | (2048, 64)|	0.8317 |
> | (4096, 32)|	0.8297 |
>
> **Q2:	For the results tables, could authors include std to show if difference is statistically significant?**
>
> - Thank you for the suggestion. In the following tables, we report the T-test between Proposed + AT and different baselines on the DNS1 test set to show the statistical significance. **!!Please check Section K in the Appendix for the colored version!!** In the tables, results shown in red, and blue represent Proposed + AT is statistically significant (p-value<0.05) better and worse than the baseline, respectively (results with black color represent no statistically significant). From the tables, we can observe that Proposed + AT is usually statistically significant better on the DNSMOS (BAK) and DNSMOS (OVR) which implies better noise removal ability. This improvement mainly comes from VQ and AT.
>
> - Table r3: P-value of DNSMOS (SIG) between Proposed + AT and baselines
> on the DNS1 test set.
> || Noisy     | Wiener | Demucs| CNN-Transformer|
> | :----:  |    :----:   |   :---  |    :----:   |   :----:   |
> | Real  |   0.683   |   0.128 |   0.0191   |   0.0025  |
> | Noreverb	|   0.743   |   2.29e-14 |   0.00126   |   0.265  |
> | Reverb	|   4.35e-24   |   3.46e-09 |   1.55e-11   |   0.0019  |
>
>
> - Table r4: P-value of DNSMOS (BAK) between Proposed + AT and baselines
> on the DNS1 test set.
> || Noisy     | Wiener | Demucs| CNN-Transformer|
> | :----:  |    :----:   |   :---  |    :----:   |   :----:   |
> | Real  |  3.77e-104	 | 3.90e-83	 | 3.53e-14	 | 3.01e-10  |
> | Noreverb	|   3.79e-61| 4.69e-49| 2.97e-08| 0.0018  |
> | Reverb	|   4.74e-91| 1.92e-46| 0.114| 0.422 |
>
>
> - Table r5: P-value of DNSMOS (OVR) between Proposed + AT and baselines
> on the DNS1 test set.
> || Noisy     | Wiener | Demucs| CNN-Transformer|
> | :----:  |    :----:   |   :---  |    :----:   |   :----:   |
> | Real  |  1.35e-36  | 	4.84e-31 | 8.97e-07	 | 0.0001  |
> | Noreverb	|   7.84e-43| 1.18e-47| 0.0042| 0.141  |
> | Reverb	|   9.20e-54| 8.70e-34| 3.21e-08| 0.205 |

---

> > ### Author Response · Authors · 2023-11-19
> > **Response to Reviewer DUwZ (Q3)**
> >
> > **Q 3: For Table 3, could authors add a short summary about comparing model complexity for the proposed approach vs. baselines?**
> >
> > - We compare model complexity based on the number of parameters and the number of computational operations as shown in Table r6. MACs stand for multiply–accumulate operation and are calculated based on 1 sec of audio input. Because NyTT doesn’t release the model, it is difficult to accurately estimate its model complexity. However, its model structure is based on CNN-BLSTM, so we can expect it to have higher model complexity compared to MetricGAN-U, which is based on simple BLSTM. CNN-Transformer is the supervised version (and removing VQ) of our proposed model and hence has a similar model complexity. Demucs is a CNN encoder and decoder framework with BLSTM in between to model temporal relationships. Because directly models the waveform, its model complexity is significantly higher than others.
> >
> > - Table r6: Model complexity for the proposed approach vs. baselines.
> > || Params (M) | MACs (G) |
> > | :----:        |    :----:   |    :----:   |
> > | CNN-Transformer	| 2.51| 0.32 |
> > | Demucs| 60.81| 75.56 |
> > | MetricGAN-U| 1.90| 0.24 |
> > | NyTT| -| - |
> > | Proposed| 2.51| 0.32 |

---

### Official Review · Reviewer_CJHD · 2023-11-10

**Soundness:** 3 good
**Presentation:** 3 good
**Contribution:** 2 fair
**Rating:** 6
**Confidence:** 3

**Summary:**

This is an interesting paper about developing a self-supervised speech enhancement solution. It does not use external noise corpus and uses a variation of VQ-VAE. It focuses on developing robust encoder and decoder using adversarial training (AT). They first train a regular VQ-VAE. Authors aptly describe the main idea as, "Once the encoder can map the noisy speech to the corresponding tokens of clean speech, or the decoder has the error correction ability, speech enhancement can be achieved." AT is then used to fine-tune encoder and decoder. Authors show high correlation of their proposed metric with other quality metrics (real+hand engineered).

**Strengths:**

1. Novelty: Authors have attempted to combine VQ-VAE and AT to create an enhancer. This is novel per my knowledge.
2. Choice of models to compare with is good.

**Weaknesses:**

1. No downstream evaluation (diarization, speaker recognition, ASR, etc.) provided which I would expect for ICLR.
2. I am not able to determine if high linear correlation of proposed metric is enough to say this enhancement will work on real noisy datasets. Remember the goal of enhancement is to remove noise (and other unwanted information) such that it can used on a plethora end applications. It is not just about perceptually making it better. STOI-like metrics are ignored in this work which quantifies intelligibility. Note that it is also possible to produce good sounding audio which is not very intelligible.
3. Lack of ablation or other analysis on proposed method. Since the proposed method is the main technical contribution, I would expect it to be evaluated more robustly.
4. Noise corpora is not used which is readily available. It would be interesting to see how using external noises can improve model performance. AT noise is not the only noise that is readily available. In fact AT is slow.
5. If TorchaudioSquim has mismatch issues (as authors point out), it can be retrained to make it more appropriate for comparison with proposed method.
6. Table 4,5 is missing PESQ, STOI numbers. (CHECK: https://paperswithcode.com/sota/speech-enhancement-on-demand). I dont understand how Weiner is best in SIG (real subset, Table 4). I am not sure dereverberation should be investigated in this paper. DNS1 details are also not mentioned.

**Questions:**

1. Why role of PESQ is downplayed? Authors say it is something to do with generative models but they did not expand or give citations to support this idea.
2. Why downstream evaluation is not done? To publish a new enhancement solution in ICLR, in my personal opinion, it becomes critical.

---

> ### Author Response · Authors · 2023-11-19
> **Response to Reviewer CJHD (Q1~Q2)**
>
> We sincerely appreciate Reviewer CJHD for considering our work as an interesting paper with novelty. Your suggestions are with decent insights for us to revise parts of the submission draft. Please find the responses below:
>
> **Q1: No downstream evaluation (diarization, speaker recognition, ASR, etc.) provided which I would expect for ICLR.**
>
> **Q8: Why downstream evaluation is not done? To publish a new enhancement solution in ICLR, in my personal opinion, it becomes critical.**
>
> - Thank you for the suggestion, we have added an ASR experimental result in Table r1 and the Appendix of the paper. We apply Whisper [r1] as the ASR model and compute the WER of speech generated by different SE models [r2] on the VoiceBank-DEMAND noisy test set. The results show that all the SE can improve the WER performance, and our proposed method can achieve the lowest WER.
>
> - Table r1: WER of speech generated by different SE models on the
> VoiceBank-DEMAND noisy test set.
> ||Noisy|Wiener|Demcus|CNN-Transformer|Proposed|
> |  :----: |  :----: |  :----: |  :----: |  :----: |  :----: |
> |Whisper ASR |14.25 |12.60 |13.75 |11.84 |**11.65**|
>
>  [r1] Radford, A., Kim, J. W., Xu, T., Brockman, G., McLeavey, C., & Sutskever, I. (2023, July). Robust speech recognition via large-scale weak supervision. In International Conference on Machine Learning (pp. 28492-28518). PMLR.
>
> [r2] Defossez, A., Synnaeve, G., & Adi, Y. (2020). Real time speech enhancement in the waveform domain. arXiv preprint arXiv:2006.12847.
>
>
> **Q 2: I am not able to determine if high linear correlation of proposed metric is enough to say this enhancement will work on real noisy datasets. Remember the goal of enhancement is to remove noise (and other unwanted information) such that it can used on a plethora end applications. It is not just about perceptually making it better. STOI-like metrics are ignored in this work which quantifies intelligibility. Note that it is also possible to produce good sounding audio which is not very intelligible.**
>
> - **For speech quality estimation:**
> We have added new results of linear correlation between STOI and our VQScore on the VoiceBank-DEMAND noisy test set in Table r2 (also added the results in Table 1 in the paper). We can observe that, for STOI, the VQScore calculated in the code space (z) still has a much larger correlation than that calculated in the signal space (x) and the self-supervised baseline, SpeechLMScore.
>
> - For real noisy datasets, our test sets, ‘Tencent_wR’ contains speech recorded in a realistic reverberant room, and ‘IUB_cosine’ dataset includes a close-talking mic (reference) and chest/shoulder mic (noisy). Therefore, real noisy datasets have already been considered in our test sets.
>
> - Table r2: LCC between STOI and our VQScore on the VoiceBank-DEMAND noisy test set.
> ||SpeechLMScore|VQScore(cos,x)|VQScore(cos,z)|
> |  :----: |  :----: |  :----: |  :----: |
> |STOI| 0.6023|0.5012|**0.7490**|
>
>
> - **For speech enhancement:**
> The reason we downplayed the role of intrusive metrics (clean reference is needed) such as PESQ, and STOI for SE evaluation is detailly explained in the response of Q7. In summary, the goal of generative models is to model the **distribution** of clean speech **rather than minimize the difference with its clean counterpart in the signal space**. Therefore, intrusive metrics may not be able to appropriately evaluate its performance.
>
> - Please refer to the response of Q7 for the discussion and related citations, etc. On the other hand, the results shown in Table r1 show that the proposed method can effectively decrease WER, which quantifies the intelligibility of an ASR.
>
> - For real noisy datasets, both DNS 1 (Table 4) and DNS3 (Table 8) test sets contain REAL noisy subsets, where our self-supervised SE model shows better performance than the supervised model, Demucs.

---

> > ### Author Response · Authors · 2023-11-19
> > **Response to Reviewer CJHD (Q3~Q4)**
> >
> > **Q3: Lack of ablation or other analysis on proposed method. Since the proposed method is the main technical contribution, I would expect it to be evaluated more robustly.**
> >
> > - **We have added several new experiments and analyses in the Appendix, including 1) Computational load of adversarial training, 2) Learning curves of adversarial training, 3) Distribution of VQScore, 4) Sensitivity to hyper-parameters, 5) Model complexity, 6) Statistical significance, and 7) ASR results of enhanced speech.**
> >
> > - An ablation study about four combinations of distance metrics and targets for calculating VQScore has already been presented in section B “COMPARISON OF DIFFERENT VQSCORES” of the Appendix. Here we present another ablation study about the codebook size. LCC between DNSMOS (OVRL) and VQScores with different codebook sizes are shown in Table r3. From this table, we can observe that except for very small codebook dimensions (i.e., 16), the performance is quite robust to codebook number and dimension. The case for speech enhancement is similar, except for very small codebook dimensions and numbers, the performance is also robust to the codebook setting. These results have been added to the Appendix of the paper.
> >
> > - Table r3: LCC between DNSMOS (OVRL) and VQScores with different codebook size.
> > |Codebook size (number, dim)|LCC|
> > |  :----: |  :----: |
> > |(1024, 32)|0.8332|
> > |(2048, 16)|0.7668|
> > |**(2048, 32)**|**0.8386**|
> > |(2048, 64)|0.8317|
> > |(4096, 32)|0.8297|
> >
> > **Q 4: Noise corpora is not used which is readily available. It would be interesting to see how using external noises can improve model performance. AT noise is not the only noise that is readily available. In fact AT is slow.**
> >
> > - Thank you for pointing this out, this is a very interesting direction to compare real noise and adversarial noise. We tried to add real noise from the DEMAND dataset just as those used in the VoiceBank-DEMAND training set. In the **dataset-matched** conditions (the training and testing sets were from the same source), adding real noise can provide **further improvement** compared to adversarial noise as shown in Table r4. On the other hand, in the **dataset-mismatched** conditions, the performance of adding real noise is **similar** to that of adding adversarial noise as shown in Table r5.
> >
> > - **About AT is slow**: Unlike the iterative-fast gradient sign method (I-FGSM) [r3] which requires an iterative procedure to generate adversarial samples, the computational cost of our AT is similar to FGSM [r4] which only needs a single pass to generate adversarial samples. Compared to normal training (such as eq (7)), our AT only needs another forward and backward pass of the computational graph (eq (5)). Therefore, the computation cost is roughly **twice** of the normal training. However, as illustrated in the learning curves on the validation set (Fig. 8) in the updated Appendix, the AT training is efficient and can converge within **5000 iterations** (within **2 hours** using a common V100 GPU). Therefore, scalability should not be a concern.
> >
> >
> > - Table r4 Comparison of adding different noises for SE model training on the VoiceBank-DEMAND noisy test set.
> > ||Training data|SIG|BAK|OVR|
> > |  :----: |  :----: | :----: |:----: |:----: |
> > |Proposed|clean speech|3.329|3.646|2.876|
> > |Proposed + AT|clean speech|3.300|3.838|2.941|
> > |Proposed + real noise|Clean speech+noise|**3.326**|**3.960**|**3.026**|
> >
> >
> > - Table r5 Comparison of adding different noises for SE model training on the DNS1 test set.
> >
> > |Real|Training data|SIG|BAK|OVR|
> > |  :----: |  :----: | :----: |:----: |:----: |
> > |  Proposed |  clean speech | 3.095 |3.365 |2.589 |
> > |Proposed + Gaussian|	clean speech	|3.152	|3.458	|2.673|
> > |Proposed + AT| clean speech| **3.156**| **3.640**| **2.750**|
> > |Proposed + real noise| Clean speech+noise| 3.142| 3.604| 2.728|
> >
> > |Noreverb|Training data|SIG|BAK|OVR|
> > |  :----: |  :----: | :----: |:----: |:----: |
> > |  Proposed |  clean speech | 3.463| 3.764| 3.066 |
> > |Proposed + Gaussian|	clean speech	|3.484 | 3.830 | 3.115|
> > |Proposed + AT| clean speech| 3.481 |  **3.960** |  **3.162** |
> > |Proposed + real noise| Clean speech+noise|  **3.485** |3.881 |3.139|
> >
> > |Reverb|Training data|SIG|BAK|OVR|
> > |  :----: |  :----: | :----: |:----: |:----: |
> > |  Proposed |  clean speech | 2.911| 3.097| 2.325 |
> > |Proposed + Gaussian|	clean speech	|2.930 | 3.222 | 2.394|
> > |Proposed + AT| clean speech| **2.949** |  3.361 |  2.456 |
> > |Proposed + real noise| Clean speech+noise|  2.859 |**3.534** |**2.465**|
> >
> > [r3] Kurakin, A., Goodfellow, I., & Bengio, S. (2016). Adversarial examples in the physical world.
> >
> > [r4] Goodfellow, I. J., Shlens, J., & Szegedy, C. (2014). Explaining and harnessing adversarial examples. arXiv preprint arXiv:1412.6572.

---

> > > ### Author Response · Authors · 2023-11-19
> > > **Response to Reviewer CJHD (Q5~Q7)**
> > >
> > > **Q5:	If TorchaudioSquim has mismatch issues (as authors point out), it can be retrained to make it more appropriate for comparison with proposed method.**
> > >
> > > - Thank you for pointing this out. Because TorchaudioSquim is a **supervised** training framework (quality labels are needed), our training data (no quality labels) cannot be used to train this model. This also shows the value and flexibility of our self-supervised VQScore.
> > >
> > > **Q6:	Table 4,5 is missing PESQ, STOI numbers. (CHECK: https://paperswithcode.com/sota/speech-enhancement-on-demand). I dont understand how Weiner is best in SIG (real subset, Table 4). I am not sure dereverberation should be investigated in this paper. DNS1 details are also not mentioned.**
> > >
> > > - Sorry for the confusion. The DNS1 dataset is the test set of the 1st Deep Noise Suppression Challenge held by Microsoft. Because the **corresponding clean speech is not released** (in the ‘real’ subset, the corresponding clean speech may even not exist), we cannot calculate PESQ/STOI scores where the **corresponding clean speech is needed** for evaluation (from which we can see the limitation of intrusive metric and why non-intrusive metrics such as VQScore is more practical).
> > >
> > >
> > > - For the traditional signal processing method such as Wiener, because of limited model ability, the result usually has less speech distortion (high SIG) but more noise remains (low BAK).
> > >
> > >
> > > **Q7: Why role of PESQ is downplayed? Authors say it is something to do with generative models but they did not expand or give citations to support this idea.**
> > >
> > > - The discussion about the problem of using PESQ to evaluate generative models is presented in **footnote 1 of page 7**: “Note that several recent papers have shown that PESQ can not reflect the true speech quality, especially for speech generated by a **generative model** (such as **GAN** Kumar et al. (2020); Liu et al. (2022), **vocoder** Du et al. (2020); Li & Yamagishi (2020); Maiti & Mandel (2020), **diffusion model** Serrà et al. (2022), etc.) We also observe that the PESQ scores of enhanced speech generated by models with VQ usually **CANNOT** reflect the true speech quality. This is mainly because the discrete tokens in VQ are shared by similar sounds, which makes the generated speech have less fidelity. However, as pointed out by DNSMOS and the following subjective listening test, this does not imply its generated speech has lower quality.”
> > >
> > > - The reason is that PESQ is an intrusive metric that is based on the comparison between the enhanced speech with its clean counterpart. On the other hand, generative models (such as GAN, VQ-VAE, diffusion model, etc.) try to model the **distribution** of real data, and hence the generated image/speech may look/sound real but the **signal distance to their clean counterpart may be large**. In summary, intrusive metrics such as PESQ, and STOI, etc. can only provide some information about signal fidelity for generative models. In the following, we list two representative examples in detail:
> > >
> > > - 1) Microsoft recently found that PESQ may not perfectly reflect the perceptual quality (please refer to Table 2 and Fig 2 in [r5]. From which they show the correlation between PESQ and Mean Opinion Score (MOS) is only **0.78** while DNSMOS can reach to **0.93**).
> > >
> > > - 2) In Meta’s ReVISE method [r6], although their method has **much better** intelligibility (WER), and quality (MOS), the low-level detail intrusive metrics (ESTOI and MCD) are **significantly worse** than the baseline (see section B.3. in their paper).
> > >
> > > [r5] Reddy, C. K., Gopal, V., & Cutler, R. (2021, June). DNSMOS: A non-intrusive perceptual objective speech quality metric to evaluate noise suppressors. IEEE International Conference on Acoustics, Speech and Signal Processing (ICASSP), 2021.
> > >
> > > [r6] Hsu, W. N., Remez, T., Shi, B., Donley, J., & Adi, Y. “Revise: Self-supervised speech resynthesis with visual input for universal and generalized speech enhancement.” CVPR, 2023.

---

> > > > ### Author Response · Authors · 2023-11-22
> > > > **Waiting for the reply**
> > > >
> > > > Dear Reviewer CJHD,
> > > >
> > > > In response to the feedback, we've conducted additional experiments to enhance the depth and robustness of our work. This includes:
> > > >
> > > > ASR results of our SE model for downstream evaluation. A more detailed discussion of why intrusive metrics (clean reference is needed) such as PESQ and STOI may not be able to reflect the performance of a generative model.
> > > >
> > > > We believe our detailed responses provide clarity on the concerns raised and are happy to have further discussion. Please let us know if you have any additional concerns before the **approaching** deadline of the Author-Reviewer discussion phase. Your feedback is pivotal to the quality of our work, and we earnestly await your thoughts.
> > > >
> > > > Thank you for your time and understanding.

---

> > > > > ### Author Response · Authors · 2023-11-23
> > > > > **Waiting for the reply -2**
> > > > >
> > > > > Dear Reviewer CJHD,
> > > > >
> > > > > We have conducted additional experiments per your suggestion to make this work more complete. This includes:
> > > > >
> > > > > 1) Downstream evaluation on ASR.
> > > > >
> > > > > 2) Investigate the correlation between STOI and VQScore.
> > > > >
> > > > > 3) Ablation study of the proposed method.
> > > > >
> > > > > 4) Comparing real noise with adversarial noise.
> > > > >
> > > > > 5) Discuss and provide more citations about why intrusive metrics such as PESQ, and STOI may not be suitable to evaluate the performance of a generative model.
> > > > >
> > > > >
> > > > > **We believe our detailed responses provide clarity on the concerns raised and are happy to have further discussion. Please let us know if you have any additional concerns before the approaching deadline of the Author-Reviewer discussion phase. Your feedback is pivotal to the quality of our work, and we earnestly await your thoughts.**
> > > > >
> > > > > Thank you for your time and understanding.

---

> > > > > > ### Comment · Reviewer_CJHD · 2023-11-23
> > > > > > **Response to authors**
> > > > > >
> > > > > > I thank the authors for addressing my concerns. It has been delightful to learn more about this topic and see the detailed results. I have therefore substantially increased my score from 3 to 6.

---

> > > > > > > ### Author Response · Authors · 2023-11-23
> > > > > > > **Response to Reviewer CJHD**
> > > > > > >
> > > > > > > Thank you very much for your positive feedback! Your comments also gave us a lot of inspiration (especially the comparison between real noise and adversarial noise) and made this manuscript better.

---

### Author Response · Authors · 2023-11-19
**General Response to All Reviewers**

We sincerely thank the efforts of all reviewers, for their valuable, professional, and constructive comments!

- We address the issues individually under each review. Given the feedback, we have made the following changes to the paper (colored in red in the revised manuscript):

- We have added several new experiments and analyses in the Appendix, including **1) Computational load of adversarial training, 2) Learning curves of adversarial training, 3) Distribution of VQScore, 4) Sensitivity to hyper-parameters, 5) Model complexity, 6) Statistical significance, and 7) ASR results of enhanced speech.**

We thank the reviewers again and look forward to any further suggestions or discussion.

---

### Meta-Review · Area_Chair_dS9P · 2023-12-01

**Metareview:**

The authors propose a new VQScore to measure speech quality, based on the quantization error of VQ-VA. Next, the authors present a two-step distillation, in which a noise component is learned such that it minimizes the performance of the quantization process, and the student model encoder is trained to invert that behavior, making it more robust to samples noisy. The decoder is trained to reduce the reconstruction error, as in any denoising approach. All reviewers agree, with different nuances,  on the novelty of VQScore and the validity of the approach. Furthermore, the experimental results support the authors' claim. The concern regarding training stability during the proposed adversarial training has not been fully resolved.

**Justification For Why Not Higher Score:**

Some aspects of the proposed technical solution require further investigation (training stability). The idea of comparing embeddings is not entirely revolutionary/novel.

**Justification For Why Not Lower Score:**

The proposed  VQscore can evaluate the quality of each invidual audio clip. The proposed solution for quality assessment and speech enhancement is sound.

---

### Decision · Program_Chairs · 2024-01-16

Accept (poster)